# The background sodium leak channel NALCN is a major controlling factor in pituitary cell excitability

Marziyeh Belal[1,2] ⓘ, Mariusz Mucha[1], Arnaud Monteil[3,4] ⓘ, Paul G. Winyard[5], Robert Pawlak[1], Jamie J. Walker[6,7,8], Joel Tabak[1] ⓘ and Mino D. C. Belle[1,9] ⓘ

[1] *University of Exeter Medical School, Hatherly Labs, Exeter, Devon, UK*
[2] *Feinberg School of Medicine, Northwestern University, Chicago, IL, USA*
[3] *IGF, University of Montpellier, CNRS, INSERM, Montpellier, France*
[4] *Department of Physiology, Faculty of Medicine Siriraj Hospital, Mahidol University, Bangkok, Thailand*
[5] *University of Exeter Medical School, Exeter, UK*
[6] *College of Engineering, Mathematics and Physical Sciences, University of Exeter, Exeter, UK*
[7] *EPSRC Centre for Predictive Modelling in Healthcare, University of Exeter, Exeter, UK*
[8] *Bristol Medical School, Translational Health Sciences, University of Bristol, Bristol, UK*
[9] *Division of Neuroscience, School of Biological Sciences, Faculty of Biology, Medicine and Health, The University of Manchester, Manchester, UK*

Handling Editors: Peying Fong & Yamuna Krishnan

The peer review history is available in the Supporting Information section of this article (https://doi.org/10.1113/JP284036#support-information-section).

**The Journal of Physiology**

**Abstract figure legend** Pituitary hormones are essential to life because they regulate important physiological processes, such as growth and development, metabolism, reproduction, and the stress response. This is achieved via signalling interplay between the brain, mainly through hypothalamic neurohormone secretion, and peripheral feedback signals that shape pituitary cell excitability. Hormonal secretion relies on the spontaneous electrical activity of pituitary cells that regulates the intracellular calcium ($[Ca^{2+}]_i$) level, an essential signalling conduit for hormonal secretion. The sodium leak channel NALCN is integral for regulating pituitary excitability by tuning cellular resting membrane potential to support spontaneous firing activity, raising $[Ca^{2+}]_i$ for hormonal secretion.

M. Belal is eligible for the Early Investigator Prize.
This article was first published as a preprint. Belal M, Mucha M, Monteil A, Winyard PG, Pawlak R, Walker JJ, Tabak J, Belle MDC. 2021. The sodium leak channel NALCN encodes the major background sodium ion conductance in mouse anterior pituitary cells. bioRxiv. https://doi.org/10.1101/2021.08.02.454810

**Abstract**  The pituitary gland produces and secretes a variety of hormones that are essential to life, such as for the regulation of growth and development, metabolism, reproduction, and the stress response. This is achieved through an intricate signalling interplay between the brain and peripheral feedback signals that shape pituitary cell excitability by regulating the ion channel properties of these cells. In addition, endocrine anterior pituitary cells spontaneously fire action potentials to regulate the intracellular calcium ($[Ca^{2+}]_i$) level, an essential signalling conduit for hormonal secretion. To this end, pituitary cells must regulate their resting membrane potential (RMP) close to the firing threshold, but the molecular identity of the ionic mechanisms responsible for this remains largely unknown. Here, we revealed that the sodium leak channel NALCN, known to modulate neuronal excitability elsewhere in the brain, regulates excitability in the mouse anterior endocrine pituitary cells. Using viral transduction combined with powerful electrophysiology methods and calcium imaging, we show that NALCN forms the major $Na^+$ leak conductance in these cells, appropriately tuning cellular RMP for sustaining spontaneous firing activity. Genetic depletion of NALCN channel activity drastically hyperpolarised these cells, suppressing their firing and $[Ca^{2+}]_i$ oscillations. Remarkably, despite this profound function of NALCN conductance in controlling pituitary cell excitability, it represents a very small fraction of the total cell conductance. Because NALCN responds to hypothalamic hormones, our results also provide a plausible mechanism through which hormonal feedback signals from the brain and body could powerfully affect pituitary activity to influence hormonal function.

(Received 5 January 2023; accepted after revision 29 October 2024; first published online 2 December 2024)

**Corresponding authors** M. D. C. Belle: Manchester, Greater Manchester, M13 9PL, UK; Marziyeh Belal: Northwestern University, Chicago, IL, USA; Joel Tabak: Hatherly Labs, Exeter, Devon, UK.     Email: Mino.Belle@manchester.ac.uk; marziyeh.belal@northwestern.edu; j.tabak@exeter.ac.uk

**Key points**

- Pituitary hormones are essential to life as they regulate important physiological processes, such as growth and development, metabolism, reproduction and the stress response.
- Pituitary hormonal secretion relies on the spontaneous electrical activity of pituitary cells and co-ordinated inputs from the brain and periphery. This appropriately regulates intracellular calcium signals in pituitary cells to trigger hormonal release.
- Using viral transduction in combination with electrophysiology and calcium imaging, we show that the activity of the background leak channel NALCN is a major controlling factor in eliciting spontaneous electrical activity and intracellular calcium signalling in pituitary cells.
- Remarkably, our results revealed that a minute change in NALCN activity could have a major influence on pituitary cell excitability.
- Our study provides a plausible mechanism through which the brain and body could intricately control pituitary activity to influence hormonal function.

**Marziyeh Belal's** has an interest in focusing on the study of ion channels and how they regulate the activity of excitable cells in the CNS in health and disease models. She was a recipient of a PhD scholarship by the University of Exeter Medical School. During her PhD, she investigated the role of non-selective cationic leak channels in regulating the spontaneous firing and cytosolic calcium activity in anterior pituitary cells. She is currently working as a Postdoctoral Researcher in Dr James Surmeier's lab at Northwestern University, investigating how striatal GABAergic and cholinergic interneuron networks regulate the membrane excitability of striatal spiny projection neurons involved in motor movements in healthy and Parkinsonian models.

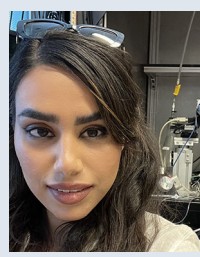

## Introduction

The electrical activity of excitable cells appropriately responds to signals that are externally driven and/or those that are intrinsically derived to support essential physiological functions, such as breathing, heartbeat and hormone release (Bertram et al., 2018; Cui et al., 2016; Protze et al., 2017; Rorsman & Ashcroft, 2018). Such excitability capacity relies on depolarising conductances that appropriately maintain the resting membrane potential (RMP) of cells near their firing threshold to support the spontaneous discharge of action potentials (APs). In hormone-secreting cells of the anterior pituitary gland, spontaneous AP firing results in rhythmic $Ca^{2+}$ entry through voltage-gated calcium channels. The resulting cytosolic/intracellular calcium ($[Ca^{2+}]_i$) oscillations serve a plethora of key physiological purposes, such as maintaining $Ca^{2+}$ levels in intracellular calcium stores, regulating gene expression and evoking hormonal secretion (Kwiecien & Hammond, 1998; Mollard & Schlegel, 1996; Stojilkovic et al., 2012). Indeed, the silencing of spontaneous firing in endocrine pituitary cells abolishes $[Ca^{2+}]_i$ oscillations and hormone secretion (Kucka et al., 2010).

The ability of pituitary cells to produce spontaneous APs is partly the result of the maintenance of their depolarised RMP relative to the $K^+$ equilibrium potential (Fletcher et al., 2018). Replacing extracellular $Na^+$ with large impermeable cations, such as $NMDG^+$, starves the cells of $Na^+$ entry, suppressing the cell's ability to depolarise. Indeed, such treatments hyperpolarise the RMP, abolishing firing activity and $[Ca^{2+}]_i$ oscillations (Kayano et al., 2019; Kucka et al., 2010, 2012; Kwiecien & Hammond, 1998; Liang et al., 2011; Sankaranarayanan & Simasko, 1996; Simasko, 1994; Tomić et al., 2011; Tsaneva-Atanasova et al., 2007; Zemkova et al., 2016). This indicates that constitutively active inward-depolarising $Na^+$-dependent currents/conductance in pituitary cells critically operates to maintain RMP close to the AP firing threshold. Pharmacological and electro-physiological investigation of this inward leak current suggested it to be mediated by a TTX-insensitive, voltage independent and constitutively active $Na^+$-permeable channel/conductance (Fletcher et al., 2018). However, the molecular identity of the channels responsible for such resting $Na^+$ conductance in pituitary cells remains unknown.

The voltage independent non-selective $Na^+$ leak channel NALCN has emerged as the major background $Na^+$-dependent conductance in several neuronal populations (Monteil et al., 2024). For example, NALCN is essential for maintaining spontaneous AP firing in hippocampal neurons (Lu et al., 2007), GABAergic and dopaminergic neurons of the midbrain (Lutas et al., 2016; Philippart & Khaliq, 2018) and neurons of the supra-chiasmatic nucleus (Flourakis et al., 2015). In the ventral respiratory neurons of the brain stem, NALCN activity facilitates rhythmic and carbon dioxide-stimulated breathing, as well as responsiveness to neuropeptides (Lu et al., 2007; Shi et al., 2016; Yeh et al., 2017), and NALCN also regulates excitability in endocrine cells, such as pancreatic $\beta$ cells (Swayne et al., 2009).

In the pituitary, there is a high level of NALCN mRNA expression (Swayne et al., 2009); accordingly, recent transcriptomic data also indicate that anterior pituitary cells express the NALCN gene and its known regulatory subunits (UNC-79, UNC-80, FAM155A) at a significantly higher level compared to other known cationic leak channels, such as the TRP and HCN channels (Paul Le Tissier, Jacques Drouin and Patrice Mollard, personal communication, April 2021). Moreover, the pharmacological profile of background $Na^+$ conductance measured in pituitary cells is similar to NALCN-mediated conductance seen elsewhere in the brain: TTX-insensitive, extracellular $Ca^{2+}$-($[Ca^{2+}]_e$) and $NMDG^+$-sensitive (Fletcher et al., 2018; Lu et al., 2007), and recent work has indicated that NALCN activity influences excitability in the GH3 pituitary clonal cell line (Impheng et al., 2021). Together, these data strongly suggest that NALCN could be the primary contributor to the background $Na^+$ conductance and $[Ca^{2+}]_i$ oscillations in anterior endocrine pituitary cells, and consequently it plays a key role in regulating pituitary cell excitability.

To evaluate this, we used a lentiviral-mediated NALCN knockdown strategy combined with powerful electro-physiology methods and calcium imaging in mouse primary anterior pituitary cells. Our results revealed that NALCN is the main contributor to the background $Na^+$-dependant depolarising conductance in pituitary cells, which, despite being a small contributor to over-all cell conductance, has a vital control over cellular excitability.

## Methods

### Animals and primary cell culture

**Ethical approval.** All experimental procedures were performed according to the provisions of the UK Animal (Scientific Procedures) Act 1986 and approved by the Research Ethics Committee of the University of Exeter (approval reference number: PP5674327). The mice were group-housed under a 12:12 h light/dark photocycle (lights on at 06.00 h), maintained at 22 $\pm$ 3°C, with food and water available *ad libitum*. Endocrine anterior pituitary cell cultures were prepared between 08.00 h and 11.00 h from wild-type male and female C57BL/6J mice (2–6 months old, bred at the University of Exeter Biological Service Facility). All animal experiments

presented in this study comply with the policies and regulations of *The Journal of Physiology* (Grundy, 2015).

For each culture preparation, three to four mice were humanely killed by cervical dislocation in accordance with a schedule 1 procedure in the absence of anaesthetics. After removing the brain, the pituitary glands were removed from the sella turcica (bony cavity) and maintained on ice at 4°C in a $100 \times 21$ mm culture dish (Thermo Fisher Scientific, Oxford, UK) containing 150 μL of Dulbecco's modified Eagle's medium (DMEM), containing high glucose and 25 mM Hepes (Sigma-Aldrich, Merck, Gillingham, UK]). Under a dissection microscope, the intermediate and posterior lobes were removed using a scalpel blade (size 10), and the anterior lobes were manually chopped into smaller pieces. Subsequently, the chopped tissues were transferred into a 50 ml falcon tube containing 2.5 mL of DMEM supplemented with 207 TAME units mL$^{-1}$ trypsin and 36 Kunitz units mL$^{-1}$ DNase I (Sigma–Aldrich) and incubated in a water bath at 37°C for 10 min. Every 5 min, the tube was swirled to prevent tissue aggregation and facilitate enzymatic digestion. After 10 min, the suspension was gently triturated 20–30 times using a 1-mL pipette tip. At the end of the digestion step, an inhibition solution containing 5 mL of DMEM, supplemented with 0.25 mg mL$^{-1}$ Lima soybean trypsin inhibitor, 100 kallikrein units of aprotinin, and 36 Kunitz units mL$^{-1}$ DNase I (Sigma-Aldrich) were added to the digestion solution. The cell suspension was then left for a few minutes to inactivate the trypsin enzyme activity. The resulting suspension was finally filtered through a cell strainer with 70 μm nylon mesh (Sigma-Aldrich, Merck) and was centrifuged at 100 *g* for 10 min. The pellet was resuspended in 500–600 μL of DMEM solution, and then 60 μL was plated on each 15 mm diameter round coverslip (Thermo Fisher Scientific) in a 12-well plate. After 20 min, once the cells were securely attached to the bottom of coverslips, 1 mL of growth medium [DMEM + 2.5% fetal bovine serum (FBS) + 0.1 % fibronectin + 1% antibiotic-penicillin/streptomycin] (Sigma–Aldrich, Merk) was added to each well and then maintained in a humidified incubator at 37°C and 5% $CO_2$. The culture medium was replaced with antibiotic-free growth-medium 6 h later. The growth medium was refreshed every 2 days.

### Lentivirus vector and cell transduction

The methods and validation used for NALCN channel knockdown (NALCN KD) were performed as described previously (Impheng et al., 2021). Briefly, a microRNA-adapted short hairpin RNA (shRNA) based on miR-30 for specific NALCN silencing integrated into a lentiviral pGIPZ plasmid that targets the 5′-GCAACAGACTGTGGCAATT-3′

region of the rat/mouse NALCN-encoding RNA was obtained commercially (Dharmacon, Lafayette, CO, USA; #V2LMM_90196). A non-silencing (NS) control (or NS Ctrl) for the NALCN channel was used in our experiments (Dharmacon; #RHS4346). The specificity of the knockdown was assayed with the use of a second sh/miR-30 RNA that targets the 5′-TTAATCCAGAGTATGTCAG-3′ (Dharmacon; #V2LMM_77139) (see Figs 2*G*, 3*E*, *F* and 6).

After 24 h in culture, 5 μL of concentrated lentivirus suspension was added to the culture media in each well of a 12-well plate containing the pituitary cells. Fresh growth medium was employed 24 h after transduction. Cells containing enhancedgreen fluorescent protein (eGFP) were usually observable 2–3 days after transduction, an indication of successful transduction and were selected for electrophysiological recordings. Only cells with high fluorescence were targeted for electrophysiology or calcium imaging. Each batch of primary pituitary cell culture was utilised for up to 5 days after transduction, both for electrophysiological recordings and calcium imaging.

### Electrophysiological recording

For electrophysiology recordings, coverslips containing pituitary cells were lowered in a recording chamber (volume ∼0.2 mL) that was fixed onto the stage of an inverted microscope (Nikon Eclipse Ti; Nikon Instruments Inc., Tokyo, Japan). Targeted recordings (current or voltage clamp) were performed from visually identified pituitary cells (at 30×) using transmitted and fluorescence lights. Transduced eGFP-positive cells were identified at 488 nm excitation using a Lambda DG-4 system (Sutter Instrument Company, Novato CA, USA). Cells with low or no fluorescence may indicate poor viral transduction and were not selected for recordings. Cells were constantly perfused at room temperature using a gravity-driven perfusion system with an extracellular solution containing (in mM) 138 NaCl, 5 KCl, 10 alpha-D-glucose, 25 Hepes, 0.7 Na$_2$HPO$_4$, 1 MgCl$_2$ and 2 CaCl$_2$ at a flow rate of ∼0.5 mL min$^{-1}$. The pH was adjusted to 7.4 with NaOH, and the osmolality was maintained at 305 mosmol L$^{-1}$. The recordings were obtained using an Axopatch 700B amplifier and Clampex 10.1 software (Molecular Devices, San Jose, CA, USA) with a sampling rate of 10 kHz, lower pass filtered at 2 kHz. Patch pipettes (borosilicate glass outer diameter: 1.5 mm; inner diameter: 0.86 mm (Warner Instrument/Multi-Channel Systems, Reutlingen, Germany) were fabricated using a horizontal Flaming/Brown micropipette puller (Sutter Instruments Company, model P-97). Pipette tips were then fire-polished and had a final tip resistance ranging from 4 to 6 MΩ.

Patch pipettes were filled with an intracellular solution containing (in mм) 10 NaCl, 100 K-gluconate, 50 KCl, 10 Hepes and 1 $MgCl_2$. The pH was adjusted to 7.2 with KOH, and osmolality at 295 mOsmol $L^{-1}$. For perforated patch recordings, 2.5 μL of amphotericin-B (stock solution at 20 mg $mL^{-1}$ in dimethyl sulfoxide) was added to 1 mL of pipette/intracellular solution to achieve a final concentration of 50 μg $mL^{-1}$.

Membrane perforation with access resistance of less than 100 MΩ was reached within 10 min after the high resistance giga-ohm seal (>10 GΩ) formation. Cells were discarded if the seal resistance was less than 10 GΩ. In the current clamp configuration, series resistance was compensated by bridge-balance and was less than 40 MΩ. Junction potential was not corrected for. In the voltage clamp configuration, series resistance was normally less than 50 MΩ and was electronically compensated (~60%). Cell capacitance ranged between 4 and 6 pF and was also compensated. All salts used for electrophysiology and Amphotericin-B were purchased from Sigma (Sigma-Aldrich, Merk).

### Dynamic clamp

To achieve the speed and precision for dynamic clamp recordings, a separate computer from the one used for current or voltage clamp data acquisition was used. The computer was connected to a National Instruments digital-to-analogue interface (DAQ; National Instruments Inc., Theale, UK) and ran the dynamic clamp module in QuB software (Milescu et al., 2008). In the current clamp configuration of the multiclamp 700B amplifier, the membrane voltage ($V_m$) of a patched cell was recorded in real-time and feed to the computer running QuB through the DAQ system as an input to calculate the current through non-selective cationic leak channels, calculated according to: $I_{NS} = g_{NS} (V_m - E_{NS})$. This defines the corresponding current ($I_{NS}$) going through the cationic leak channels. The sodium leak conductance ($g_{NS}$) was then manipulated in QuB, and the calculated $I_{NS}$ was then injected back to the cell in real time via the same DAQ. Responses in cellular membrane voltage ($V_m$) were then recorded in pClamp. In this configuration, the injected NALCN-like current is dynamic, unlike during conventional current clamp recordings where the injected current remains constant at all time points.

The reversal potential ($E_{NS}$) for the channels was set at zero, corresponding to values for non-selective cationic leak channels (Chua et al., 2020; Lu et al., 2007).

### Measurement of cytosolic calcium in single pituitary cells

The coverslips with pituitary cells were bathed in extracellular solution (containing the same ingredients as described in the electrophysiology section) containing 2 μм fura-2 AM; 1 mм stock made on the day of the experiment in 20% Pluronic F-127 in dimethyl sulfoxide (Thermo Fisher Scientific; F1221) for 45 min at 37°C. The cells were then rinsed three times with extracellular solution using a 2 mL Pasteur pipette. The coverslips were then lowered in a recording chamber (volume ~0.2 mL) that was fixed onto the stage of an inverted Nikon microscope (Nikon Eclipse Ti; Nikon Instruments Inc.). This is the same setup used in our electrophysiology experiments. Cells were constantly perfused at ~0.5 mL $min^{-1}$ with an extracellular solution at room temperature using a gravity-driven perfusion system. The Fura-2 loaded cells were excited at 340 and 380 nm wavelengths (20 ms exposure time) using a Lambda DG-4 system (Sutter Instrument Company) and imaged at 30× with a Nikon lens (Nikon Instruments Inc.). Pairs of images were captured every second. Light intensity was reduced by 50% before hitting the cells using an appropriate filter. The intensity of emitted light was measured at 520 nm by a Hamamatsu CMOS camera C1344 using $4 \times 4$ binning (Hamamatsu, Photonics, Hamamatsu City, Japan). Hardware control was achieved by TI Workbench software developed by T. Inoue (Inoue 2018).

For analysis, regions of interest (ROI) were selected around the cells. Care was taken to choose single cells only. Cells that had overlapping somas were not used for analysis. To measure background fluorescence, a single ROI was selected in an area that contained no cells. Pixel intensity values within each region of interest for both 340 and 380 nm excitation wavelengths were averaged and then subtracted from the background fluorescence. Ratiometric value ($r$) was then computed for each cell according:

$$r = \left( ROI_{340} - ROI_{background340} \right) / \left( ROI_{380} - ROI_{background380} \right).$$

Analysis was performed in MATLAB software (MathWorks, Inc., Natick, MA, USA).

### HEK-293T cell culture

HEK-293T cells were obtained from the European Collection of Authenticated Cell Cultures (ECACC, Porton Down, UK; #96121229). The identity of HEK-293T has been confirmed by STR profiling and the cells have been eradicated from mycoplasma at ECACC. We routinely tested the cells for the absence of any mycoplasma contamination. Cells were cultivated at 37°C in DMEM supplemented with GlutaMax (Thermo Fisher Scientific; #31966047), 10% fetal bovine serum (Thermo Fisher Scientific; #10270106) and 1% penicillin/streptomycin (Thermo Fisher Scientific; #15140122).

## Plasmids and transfection

pcDNA3-NALCN-GFP, pcDNA3-UNC-79, pcDNA3-UNC-80 and pcDNA3-FAM155A plasmids were kindly provided by Professor Stephan Pless and Dr Han Chua (Chua et al., 2020). Transfections of HEK-293T cells were performed with cells cultured on poly-L-ornithine-treated coverslips in 35 mm Petri dishes using the jetPEI reagent (Polyplus, Illkirch, France; #101000053) with 2 μg of a DNA mix containing pcDNA3-NALCN-GFP, pcDNA3-UNC-79, pcDNA3-UNC-80 and pcDNA3-FAM155A in a 1:1:1:1 molar ratio in accordance with the manufacturer's protocol.

## Immunohistochemistry

Pituitary glands were extracted and fixed overnight in 4% paraformaldehyde prepared in phosphate-buffered saline (PBST). The following day, 70 μm thick sections were cut using a vibratome (Campden Instruments, Loughborough, UK), placed on poly-L-lysine coated microscope slides (VWR, Radnor, PA, USA) and left to dry. Next, the sections were blocked with 10% FBS in PBST (PBS containing 0.01% Triton X-100) for 1 h at room temperature. Primary rabbit anti-NALCN antibody at 1:500 dilution (Alomone Labs, Jerusalem, Israel; #ASC-022) in 10% FBS/PBST was applied to the sections and incubated in a humid chamber at 4°C overnight. The next day, sections were washed three times for 15 min in PBST. Then, the secondary 488-Alexa-Fluor conjugated antibody (dilution 1:1000; Molecular Probes, Eugene, OR, USA) was applied in 10% FBS/PBST solution for 1 h at room temperature, followed by a series of three 15 min washes in PBST. The nucleic acids present in the pituitary gland cells were visualised with TOTO-3 (dilution 1:2000; Thermo Fisher Scientific). Sections were incubated with TOTO-3 for 15 min, then washed three times (3 × 10 min) and subsequently mounted with Fluorsave medium (Calbiochem, San Diego, CA, USA). In controls, primary antisera were omitted from the incubating solution. Images were captured using a LSM 5 confocal microscope with Zen software (Zeiss, Oberkochen, Germany).

For HEK-293T cells, cells were blocked with 10% FBS in PBST and then incubated with primary rabbit anti-NALCN antibody at 1:500 dilution (Alomone Labs; #ASC-022) in 10% FBS/PBST for 1 h at room temperature. Cells were washed three times for 15 min in PBST. Then, the secondary 488-Alexa-Fluor conjugated antibody (dilution 1:1000; Molecular Probes) was applied in 10% FBS/PBST solution for 1 h at room temperature, followed by a series of three 15 min washes in PBST. Then, the secondary 594-Alexa-Fluor conjugated antibody (dilution 1:1000; Molecular Probes) was applied in 10% FBS/PBST solution for 1 h at room temperature, followed by a series of three 15 min washes in PBST. Images were captured using a ApoTome microscope (Zeiss) and then analysed with Fiji software (Schindelin et al., 2012).

## Statistical analysis

MATLAB software (MathWorks, Inc.) was used for all statistical analysis, and the type of statistical test used in each case is specified in the Results section, as appropriate.

# Results

## NALCN channel protein is expressed in primary mouse anterior pituitary cells

Previous research has revealed the expression of the *Nalcn* gene in mouse pituitary gland at the mRNA level (Swayne et al., 2009). Thus, we first aimed to localise NALCN expression in mouse primary endocrine pituitary cells using a commercially available NALCN antibody (Alomone Labs; #ASC-022; Li et al., 2021; Zhang et al., 2021). Because the specificity of this NALCN antibody is not clearly established, we performed NALCN immunofluorescence (NALCN-IF) in HEK293T cells transfected with NALCN-GFP with its subunits (NALCN + UNC-79 + UNC-80 + FAM155A) as a control. We observed colocalisation of NALCN-IF and NALCN-GFP signals, suggesting some NALCN-IF antibody specificity with NALCN-GFP+ve cells (Fig. 1). We next performed NALCN-IF on pituitary slices. Our results revealed the presence of NALCN-IF (Fig. 1), with the majority of the anterior pituitary cells stained for NALCN. In addition, the cellular localisation of the NALCN-immunofluorescence is consistent with previous observations (Li et al., 2021; Zhang et al., 2021).

## NALCN channel knockdown silences electrical activity in pituitary cells

Several studies have shown that functional non-selective cationic channels (NSCC) are responsible for excitability in pituitary cells (Fletcher et al., 2018; Stojilkovic et al., 2010), but the identity of the channels responsible remains elusive. To determine whether NALCN is the main NSCC in endocrine anterior pituitary cells, we used a genetic manipulation approach to directly and selectively alter the activity of NALCN, investigating its role in the regulation of membrane potential and spontaneous firing in these cells. To this end, we used a lentiviral-mediated knockdown strategy (a shRNA that targets the 5′-GCAACAGACTGTGGCAATT-3′ region of the rat/mouse NALCN-encoding RNA,

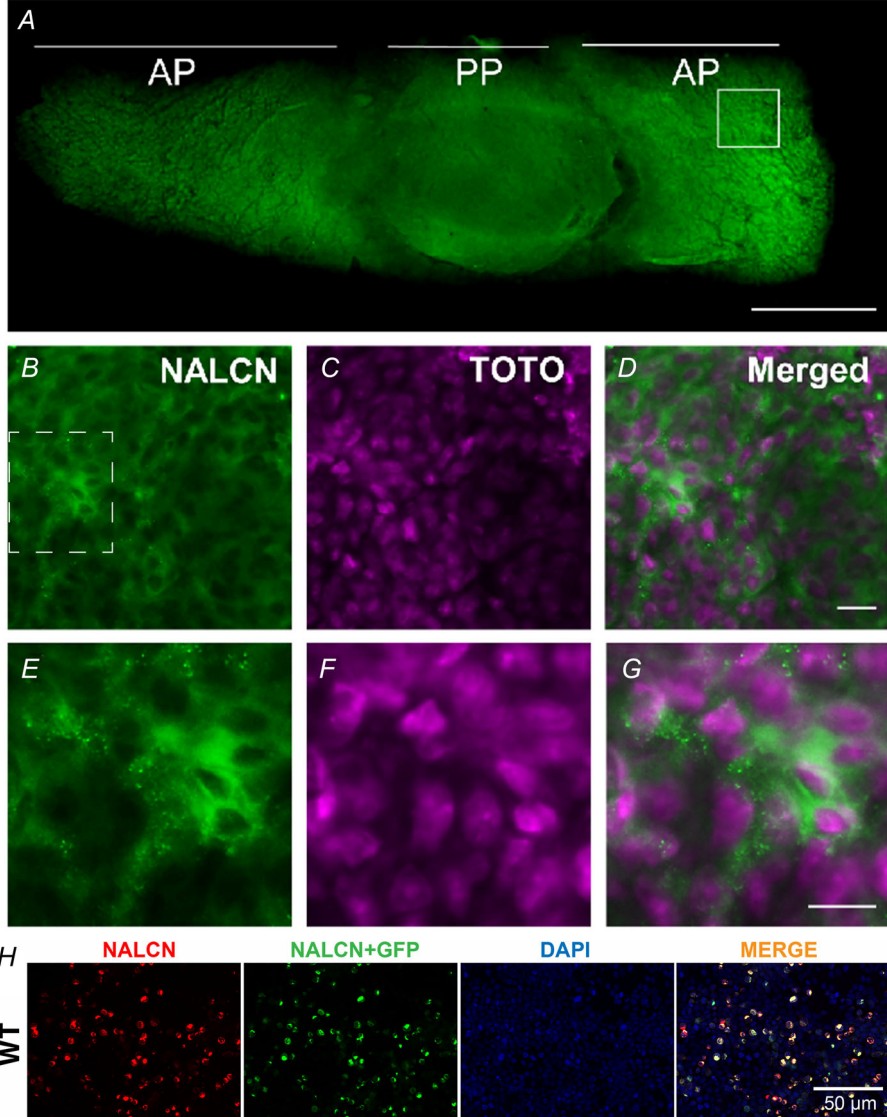

**Figure 1. The mouse pituitary gland expresses the NALCN ion channel**
Immunofluorescence staining revealed the presence of NALCN channel protein in the mouse anterior pituitary gland (*n* = 3 animals). The NALCN channel is shown in green. TOTO-3 (shown in magenta) was used to identify cell bodies by visualising cellular nucleic acids. *A*, transverse section of a pituitary gland under 10× magnification excited with 488 nm light to detect NALCN channel immunofluorescence. *B*, NALCN channel visualisation at 20× taken in an area within the white solid square in (*A*). *C*, TOTO-labelled nucleic acid in the same area shown in (*B*). *D*, merged image from (*B*) and (*C*). *B–D*, 20× magnification. *E–G*, insets under higher digital magnification using confocal microscopy from the area defined by the white dashed square in (*B*). Scale bars = 500 μm in (*A*) and 20 μm in (*D*) and (*G*). AP, anterior pituitary; PP, posterior pituitary, respectively. *H*, NALCN immunofluorescence (NALCN-IF) colocalises with transfected NALCN-GFP in HEK cells. Showing NALCN-IF in HEK293T cells 48 h following transfection with NALCN-GFP along with its ancillary subunits at a 1:1:1:1 molar ratio (NALCN + UNC-79 + UNC-80 + FAM155A). Colocalisation of NALCN-GFP and NALCN-IF signals can be seen. We performed two experimental replicates and obtained similar results, providing some support for the specificity of the NALCN antibody used in our main study (Alomone Labs; #ASC-022). NALCN-IF in red; NALCN-GFP in green; DAPI in blue and merged signals. Scale bar = 50 μm. Note that NALCN channel protein is mostly detected intracellularly although a NALCN current is recorded using the patch clamp technique with this configuration (data not shown). [Colour figure can be viewed at wileyonlinelibrary.com]

and a second sh/miR-30 RNA that targets the 5′-TTAATCCAGAGTATGTCAG-3′ region to validate the specificity of the knockdown) to decrease NALCN channel activity in mouse endocrine pituitary cells (see Methods). We then performed targeted electrophysiology recordings on visually identified pituitary cells to evaluate the resulting changes in the electrophysiological properties of these cells.

Our data revealed that almost all untreated control (untreated Ctrl) and non-silencing control (NS Ctrl, targeted by presence of eGFP) cells exhibited spontaneous firing activity (Fig. 2*A*, *B*, *D* and *F*), consistent with previous reports (Fletcher et al., 2018). By contrast, 90% (28 of 31 cells from 12 animals) of NALCN knockdown (NALCN KD, targeted by presence of eGFP) cells were silent (Fig. 2*C*, *D* and *F*), compared to just 17% (five of 30 cells from 12 animals) in NS Ctrl and 19% (six of 31 cells from 12 animals) in untreated control cells (Fig. 2*D* and *F*). In addition, NALCN KD cells exhibited a significantly more hyperpolarised RMP compared to both untreated and NS Ctrl cells (NALCN KD RMP: −63.6 ± 8.5 mV, $n = 31$ cells, from 12 animals; NS Ctrl: −45.04 ± 5.4 mV, $n = 30$ cells, from 12 animals; untreated Ctrl: −47.9 ± 7.3 mV, $n = 31$ cells, from 12 animals; $P = 0.00034$, one-way

ANOVA) (Fig. 2*E*). NS Ctrl and untreated Ctrl cells had similar firing frequency (untreated Ctrl: 0.50 ± 0.4 Hz, $n = 31$ cells; NS Ctrl: 0.49 ± 0.4 Hz, $n = 30$ cells, $P = 0.8$, $t$ test; from 12 animals in each condition) (Fig. 2*F*) and RMP (untreated Ctrl: −47.9 ± 7.3 mV, $n = 31$ cells; NS Ctrl: −45.04 mV ± 5.4 mV, $n = 30$ cells, $P = 0.086$, $t$ test; from 12 animals in each condition) (Fig. 2*E*), suggesting that non-silencing NS Ctrl had no effect on firing activity or RMP of pituitary cells. To provide additional validation support for the NALCN knockdown approach used, we repeated our experiments with a separate shRNA (SH2-RNA or sh/miR-30) that targets a different sequence (5′-TTAATCCAGAGTATGTCAG-3′) of mouse NALCN channels. GFP-targeted patch clamp recordings from these cells supported our previous observations, with similar cellular excitability suppression following SH2-RNA NALCN knockdown (Fig. 2*G*; see also Figs 3*E* and *F* and 6). Indeed, SH2-RNA-mediated NALCN KD cells were significantly hyperpolarised-silent when compared with untreated cells (NALCN KD RMP: −64 mV ± 5 mV, $n = 7$ cells, from three animals; untreated Ctrl: −43.75 ± 6.75 mV, $n = 20$ cells, from eight animals; $P = 0.0004$; Mann–Whitney test) (Fig. 2*G*). Our results therefore indicate that NALCN activity in pituitary

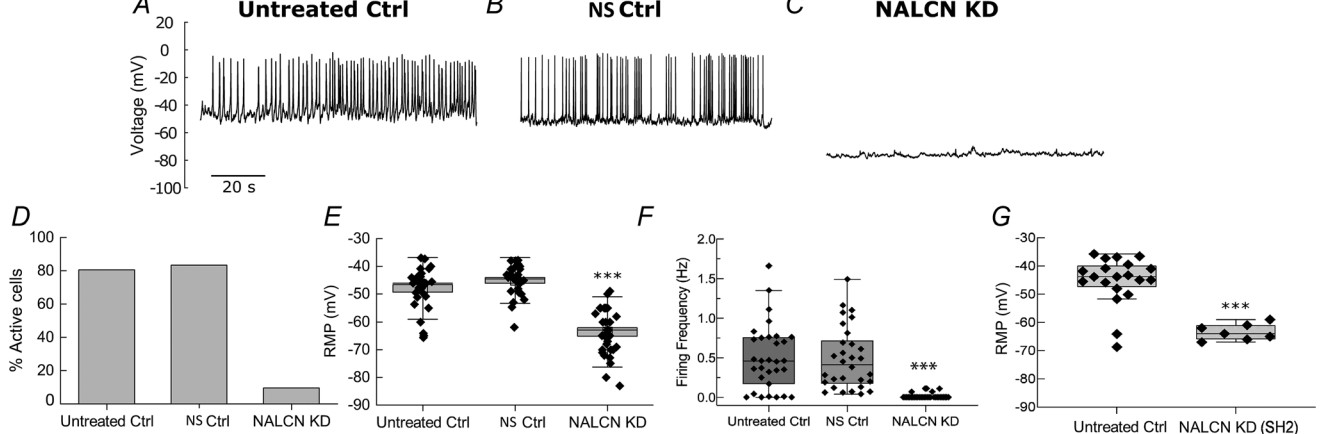

**Figure 2. Knockdown of NALCN silences electrical activity in mouse pituitary cells**
Representative traces of spontaneous firing activity from an untreated (*A*) and a non-silencing control (NS Ctrl) (*B*) pituitary cell. *C*, representative trace of electrical activity in a NALCN knockdown (NALCN KD) pituitary cell. *D*, overall percentages of active and silent cells in untreated control, NS Ctrl and NALCN KD pituitary cells. *E*, resting membrane potential of untreated control, NS Ctrl and NALCN KD pituitary cells (untreated control: −47.9 ± 7.3 mV, $n = 31$ cells *vs.* NS Ctrl: −45.04 mV ± 5.4 mV, $n = 30$ cells, from 12 animals in each condition, $P = 0.086$; NALCN KD: −63.6 ± 8.5 mV, $n = 31$ cells *vs.* NS Ctrl and untreated control, $n = 30$ cells, from 12 animals in each condition $P = 0.00034$, one-way ANOVA with Bonferroni correction). *F*, distribution of firing frequency between untreated Ctrl, NS Ctrl and NALCN KD cells over a course of 600 s (NALCN KD: 0 Hz, $n = 31$ *vs.* NS Ctrl: 0.49 ± 0.4 Hz, $n = 30$ and *vs.* untreated Ctrl: 0.50 ± 0.4 Hz, $n = 31$, $P = 0.00063$, one-way ANOVA with Bonferroni correction, from 12 animals in each condition; NS Ctrl *vs.* untreated Ctrl: $P = 0.8$, $t$ test). Only three of 31 NALCN KD cells fired action potential, albeit at a lower frequency (0.1 ± 0.02 Hz) relative to the untreated and NS Ctrl. Data are represented as the mean ± SD. *G*, validation experiments showing the resting membrane potential of untreated Ctrl cells and SH2-RNA NALCN knockdown cells (untreated Ctrl: −43.75 ± 6.75 mV, $n = 20$ cells *vs.* NALCN KD (SH2): −64 mV ± 5 mV, $n = 7$ cells, from eight and three animals, respectively, $P = 0.0004$; Mann–Whitney test). In (*E*) and (*F*), boxes and whiskers represent the SE of the mean and SD, respectively. In (*G*), boxes and whiskers represent interquartile and range, respectively. Each black dotes in (*E*) to (*G*) shows values from individual cells. ***$P < 0.001$; the *y*-axis scale bar in (*A*) also applies to (*B*) and (*C*).

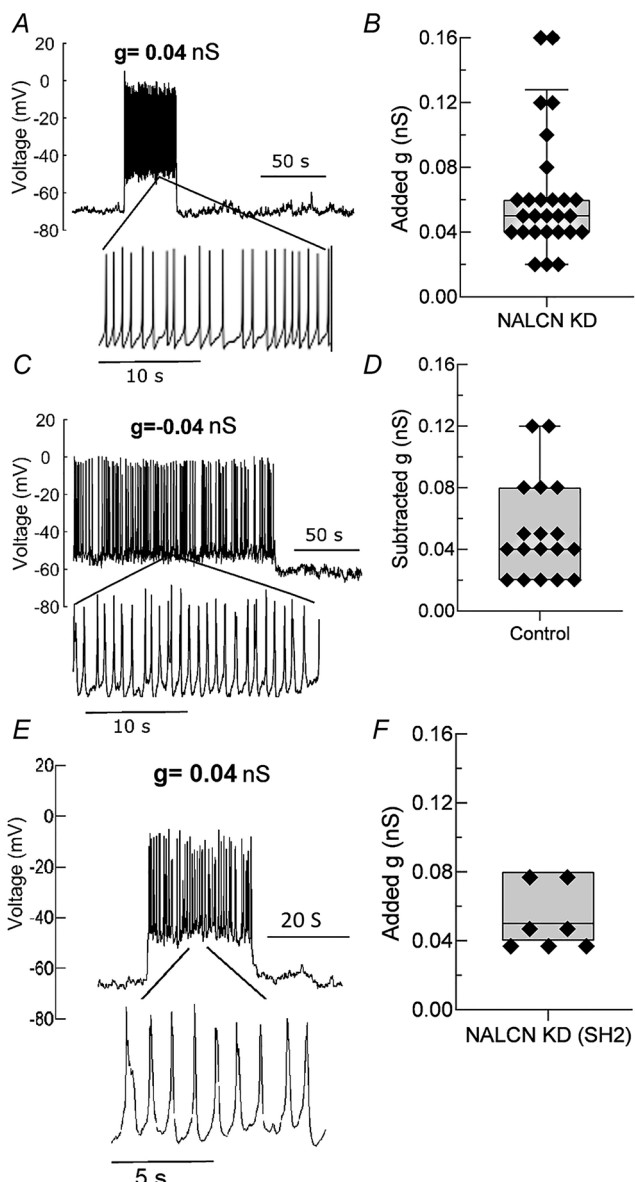

were compared. *E*, a representative voltage trace from a silent SH2-RNA-mediated NALCN KD cell recorded during knockdown validation (see Methods), with RMP and firing activity being restored once the non-selective cationic conductance (g) was injected into the cells (note the small conductance, 0.04 nS, was sufficient to restore RMP and firing. *F*, the distribution of added conductance (g) values required for restoring RMP and firing activity in hyperpolarised-silent SH2-RNA NALCN KD cells (median: 0.05 nS, *n* = 7 cells from three animals).

cells is critical for maintaining both firing activity and RMP. In addition, because our results strongly indicated that the viral transduction did not affect the firing activity or RMP of pituitary cells *per se*, we made comparisons between NALCN KD and NS Ctrl cells only.

## Minute conductance injection restores firing in NALCN KD primary pituitary cells

Having rendered pituitary cells hyperpolarised-silent with NALCN KD, we next used dynamic clamp to examine whether the injection of non-selective cationic conductance to these silent NALCN KD cells could restore RMP and spontaneous firing activity. Indeed, hyperpolarised-silent NALCN KD cells became depolarised and restored typical firing activity and frequency when the non-selective cationic conductance was injected at small values, even as low as 0.02 nS (Fig. 3*A* and *B*). Overall, the added conductance required to bring NALCN KD cells to the typical RMP and firing activity/frequency observed in NS Ctrl cells ranged from 0.02 to 0.16 nS (Fig. 3*B*). Remarkably, this was equivalent to the conductance range required (−0.02 to −0.12 nS) to silence spontaneous activity in NS Ctrl cells (Fig. 3*C* and *D*). For knockdown validation control, we then repeated the dynamic clamp stimulation experiments in our hyperpolarised-silent SH2-RNA-mediated NALCN KD cells and found that a similar range of applied conductance (0.04 to 0.8 nS) restored depolarised RMP and firing in these cells (Fig. 3*E* and *F*; see also Figs 2*G* and 6).

Taken together, this first set of results indicates that NALCN is a key player in pituitary cell excitability through its modulatory effect on RMP and, subsequently, firing activity in primary pituitary cells. The magnitude of NALCN conductance lost by the cells after NALCN KD, as determined by the dynamic clamp, was estimated to be in the order of 0.05 nS.

## NALCN majorly contributes to the inward Na+ leak currents in primary pituitary cells

To further determine the contribution of the NALCN-mediated inward leak current over other back-

**Figure 3. Dynamic clamp addition of non-selective cationic conductance restores firing activity in silent NALCN KD pituitary cells**

*A*, a representative voltage trace of silent NALCN KD pituitary cells with firing being restored. Firing activity was restored once the non-selective cationic conductance (g) was increased by an amount as minute as 0.04 nS (e.g. silent cells depolarised and discharged barrages of action potentials when 0.04 nS conductance was applied, immediately returning to baseline after termination of added conductance). *B*, the distribution of added conductance (g) values required for restoring firing activity in hyperpolarised-silent NALCN KD pituitary cells. Median: 0.05 nS, *n* = 27 cells from 12 animals. *C*, a control cell firing spontaneously. Conversely, dynamic clamp subtraction/removal of 0.04 nS of non-selective cationic conductance silenced the cell (compare with *B*). *D*, the distribution of subtracted conductance (g) values required for silencing firing activity in spontaneously active pituitary cells (median: 0.04 nS, *n* = 18 cells from 10 animals). Note that there is no significant difference (*P* = 0.92, *t* test) between the two groups when firing frequency and resting membrane potential of cells in (*A*) and (*C*)

ground Na$^+$ currents in pituitary cells, we compared the inward current density in NS Ctrl and NALCN KD cells when maintaining them at a holding potential of $-80$ mV, which also minimised any influence of K$^+$ channel conductance. At $-80$ mV, the holding current density was significantly larger in the NS Ctrl group compared to the NALCN KD group (NS Ctrl: $-0.72 \pm 0.2$ pA/pF, $n = 14$ cells, from eight animals; NALCN KD: $-0.24 \pm 0.13$ pA/pF, $n = 16$ cells, from eight animals; $P = 0.00021$, $t$ test) (Fig. 4$A$–$C$), confirming that the NALCN KD cells were indeed more hyperpolarised than their NS Ctrl counterparts. Remarkably, this background Na$^+$ conductance was reduced by 0.045 nS in NS Ctrl, but by only 0.015 nS in NALCN KD (three times smaller than that measured in NS Ctrl) when extracellular

Na$^+$ was replaced by the Na$^+$ channel impermeant cation NMDG$^+$. This indicates that the measured inward currents in NS Ctrl were being mediated by a background Na$^+$ conductance that was almost absent in NALCN KD cells (NS Ctrl: $0.045 \pm 0.02$ nS, $n = 14$ cells; *vs.* NALCN KD: $0.015 \pm 0.01$ nS, $n = 16$ cells; from eight animals in each condition, $P = 0.00018$, Mann–Whitney test) (Fig. 4$D$).

These findings indicate that NALCN indeed contributes to most of the inward leak conductance in pituitary cells. Our results also provide an approximate estimate of NALCN conductance magnitude in pituitary cells, by noting that the difference in background Na$^+$ conductance between NS Ctrl and NALCN KD cells is 0.03 nS (0.045–0.015 nS). This is probably an underestimate because some residual NALCN conductance may still be available in NALCN KD cells following the lentiviral knockdown. Importantly, our results identify that a change in background Na$^+$ conductance of near 0.03 nS was sufficient to profoundly silence the electrical activity of endocrine anterior pituitary cell. This finding is consistent with endocrine pituitary cells having very large input resistance ($\sim$5 G$\Omega$; Dubinsky & Oxford, 1984), meaning that very small variation in the current results in drastic changes in the RMP. This also agrees with our dynamic clamp observation where a 0.05 nS (median value) inward leak conductance injection was sufficient to fully restore RMP and firing activity in NALCN KD cells (see above and also compare Figs 3 with 4).

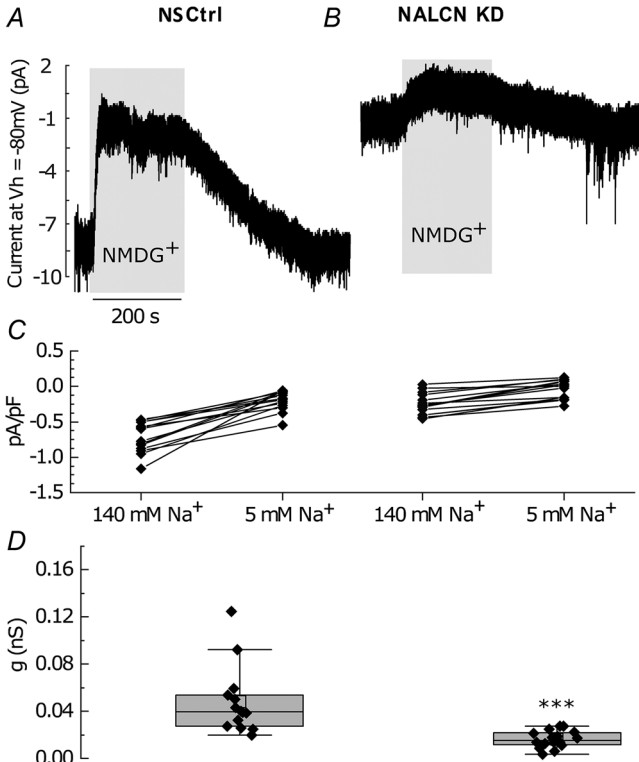

**Figure 4. NALCN contributes to most of the inward Na$^+$ leak conductance in endocrine anterior pituitary cells**
*A* and *B*, representative traces of inward Na$^+$ leak currents ($V_h = -80$ mV) in a NS Ctrl and NALCN KD pituitary cells as revealed by the replacement of extracellular Na$^+$ with NMDG$^+$ (grey boxes). *C*, the density of the inward Na$^+$ leak current in NALCN KD cells was significantly reduced compared to NS Ctrl (data presented as $\Delta$mean $\pm$ SD, NS Ctrl: $-0.72 \pm 0.2$ pA/pF ($-5.2 \pm 1.9$ pA), $n = 14$ cells; NALCN KD: $-0.24 \pm 0.13$ pA/pF ($-1.5 \pm 0.9$ pA), $n = 16$ cells; from eight animals in each condition, $P = 0.00021$, $t$ test). *D*, extracellular Na$^+$ replacement with NMDG$^+$ caused significant loss of background Na$^+$ conductance in control but not in NALCN KD cells (data represented as $\Delta$median $\pm$ interquartile, NS Ctrl: $0.045 \pm 0.02$ nS, $n = 14$; NALCN KD: $0.015 \pm 0.01$ nS, $n = 16$; $P = 0.00018$, Mann–Whitney test, eight animals in each condition).

## NALCN is required for spontaneous intracellular Ca$^{2+}$ oscillations in primary pituitary cells

Previous studies have shown that the pattern of spontaneous firing determines the amplitude and duration of [Ca$^{2+}$]$_i$ oscillations in anterior endocrine pituitary cells (Stojilkovic et al., 2005, 2012), a necessary signalling link with excitability for hormonal secretion. Thus, we next tested whether NALCN KD affects spontaneous [Ca$^{2+}$]$_i$, oscillations in these cells. We found that $\sim$57% (51/90) of NS Ctrl pituitary cells exhibited spontaneous [Ca$^{2+}$]$_i$, oscillations (Fig. 5$A$). By contrast, only 11% (4/36) of NALCN KD cells displayed such [Ca$^{2+}$]$_i$, transients (Fig. 5), whereas the remaining 89% (32/36) were quiescent and did not generate any [Ca$^{2+}$]$_i$, oscillations (Fig. 5$B$). The percentage (57%) of pituitary cells that produced [Ca$^{2+}$]$_i$, oscillations in Ctrl conditions was consistent with previous reports (Tomić et al., 2011). To quantitatively compare the size of the [Ca$^{2+}$]$_i$, oscillations between the NS Ctrl and NALCN KD cells, the SD of the baseline [Ca$^{2+}$]$_i$, ratio trace for each individual cell was calculated over 600 s. Here, we use the size of the SD as a measure of the [Ca$^{2+}$]$_i$, oscillations/fluctuations magnitude (e.g. compare Fig. 5$A$

and B). Comparing the SD between the two groups revealed a significant difference between NS Ctrl and NALCN KD cells (NS Ctrl: median = 0.04, $n$ = 90 cells; NALCN KD: median = 0.015, $n$ = 36 cells; $P$ = 0.0004, Mann–Whitney test, from five animals per condition with each animal providing roughly equal number of cells) (Fig. 5*C*), with $[Ca^{2+}]_i$, ratios showing larger fluctuations in NS Ctrl cells compared to NALCN KD cells. Together, this indicated that NALCN plays a key role in regulating $[Ca^{2+}]_i$, oscillations in primary pituitary cells. As in NS Ctrl cells, challenging NALCN KD cells with high extracellular $K^+$ (15 mм) induced depolarisation and consequently a rise in $[Ca^{2+}]_i$, ratios, serving as a positive control for these cells (Fig. 5*B*). Although we did not directly measure $V_m$ during the 15 mм $K^+$ application, the theoretical depolarisation estimated with the Nernst equation (temperature: 293 K, Z:1, $[K^+]$in: 150 mм, $[K^+]$ out: 15 mм) was −58 mV; a $V_m$ that is well within the membrane potential range for inducing firing in these cells (e.g. see Fig. 2*A* and *B*).

### NALCN mediates a low extracellular Ca²⁺-induced depolarisation in primary pituitary cells

Removal of extracellular $Ca^{2+}$ ($[Ca^{2+}]_e$) activates a low and sustained excitatory inward leak current in hippocampal neurons (Chu et al., 2003), by a mechanism that directly involves NALCN, acting through the UNC80-NALCN complex to regulate excitability (Lu et al., 2010). In primary anterior pituitary cells, lowering or elimination of $[Ca^{2+}]_e$ evoked similar effects on cellular excitability, causing severe membrane depolarisation and silencing of firing, presumably through depolarisation blockade (Kwiecien & Hammond, 1998; Sankaranarayanan & Simasko, 1996; Stojilkovic, 2006;

Tsaneva-Atanasova et al., 2007). This raises the possibility that, as in hippocampal neurons, NALCN could be responsible for the low $[Ca^{2+}]_e$-induced depolarisation in the pituitary, providing us with yet another opportunity to functionally identify NALCN in pituitary cells. We test this by assessing the effects of reducing $[Ca^{2+}]_e$ on pituitary cells excitability.

Consistent with previous studies in neurons, lowering of $[Ca^{2+}]_e$ (from 2 to 0.1 mм) caused a significant increase in inward leak current in NS Ctrl pituitary cells (median ± interquartile: −0.85 ± 0.6 pA/pF to −1.5 ± 0.5 pA/pF, $n$ = 14 cells; $P$ = 0.00049, Kruskal–Wallis, from six animals) (Fig. 6*A*, *D* and *E*). Subsequent replacement of extracellular $Na^+$ with $NMDG^+$ reduced the holding current (from −1.5 ± 0.5 pA/pF to −0.3 ± 0.2 pA/pF, $n$ = 14 cells, $P$ = 0.00053, Kruskal–Wallis) (Fig. 6*A*, *D* and *E*), indicating that the low $[Ca^{2+}]_e$ -induced depolarisation is $Na^+$ mediated. Lowering $[Ca^{2+}]_e$ in NALCN KD pituitary cells produced a rise in inward leak current (from −0.37 ± 0.35 pA/pF to −0.5 ± 0.25 pA/pF, $n$ = 13 cells, $P$ = 0.028, Kruskal–Wallis, from six animals) (Fig. 6*B*, *D* and *E*). However, in contrast, this rise in inward leak current in NALCN KD cells was substantially reduced compared to measurements in Ctrl cells (median of ∆current-density ± interquartile in NS Ctrl: −0.6 ± 0.6 pA/pF, $n$ = 14 cells, NALCN KD: −0.17 ± 0.1 pA/pF, $n$ = 13 cells, Mann–Whitney test, from six animals, $P$ = 0.00071) (Fig. 6*B*, *D* and *E*). SH2-RNA-mediated NALCN KD produced similar results where a reduction in $[Ca^{2+}]_e$ did not evoke an inward leak current in these cells (median of ∆current-density ± interquartile in SH2-RNA-NALCN KD: −0.025 ± 0.08 pA/pF, $n$ = 4 cells from three animals) (Fig. 6*C*, *D* and *E*).

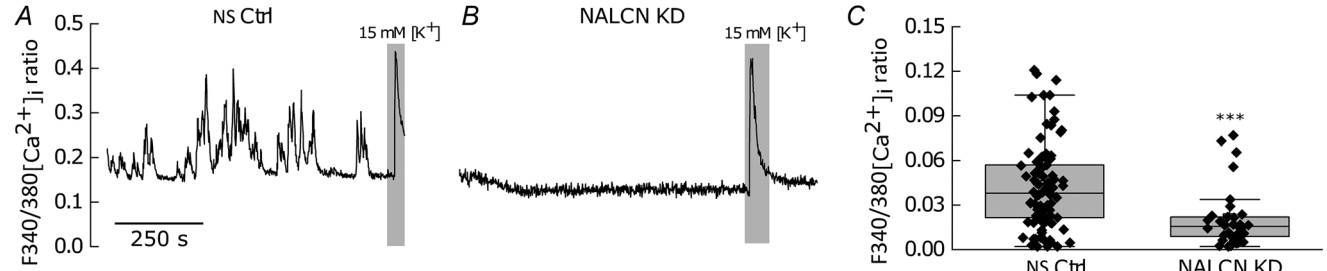

**Figure 5. NALCN KD impacts intracellular Ca²⁺ transients**
*A*, representative trace of ($[Ca^{2+}]_i$) oscillations in an NS Ctrl (NS Ctrl) pituitary cell. *B*, a representative trace of $[Ca^{2+}]_i$ in a NALCN KD pituitary cell, showing abolished $[Ca^{2+}]_i$ oscillations. *C*, the SD of each $[Ca^{2+}]_i$ ratio trace was calculated and statistical analysis revealed a significant alteration of $[Ca^{2+}]_i$ oscillations between the two groups (NS Ctrl: median = 0.04, $n$ = 90 cells; NALCN KD: median = 0.015, $n$ = 36 cells; $P$ = 0.0004, Mann–Whitney test, from five animals in each condition). Box: interquartile, Whiskers: range, excluding outliers, Central line: median. The last responses in (*A*) and (*B*) (grey boxes) represent $[Ca^{2+}]_i$ response to a rise in extracellular $K^+$ (15 mм), used here as positive control to report cell viability. Vm depolarisation was not directly measured, but theoretical depolarisation achieved with 15 mм $K^+$ was estimated with the Nernst equation to be −58 mV. The *y*-axis scale in (*A*) also applies to (*B*).

This indicates that, similar to that observed in hippocampal neurons, NALCN is involved in pituitary cell sensitivity to reduced $[Ca^{2+}]_e$ and this is strongly suggested as being the mechanism involved in the low $Ca^{2+}$-induced depolarisation observed in pituitary cells.

## Discussion

We have used a lentiviral-mediated knockdown approach combined with powerful electrophysiology methods and calcium imaging to provide the first compelling evidence that NALCN forms the dominant depolarising

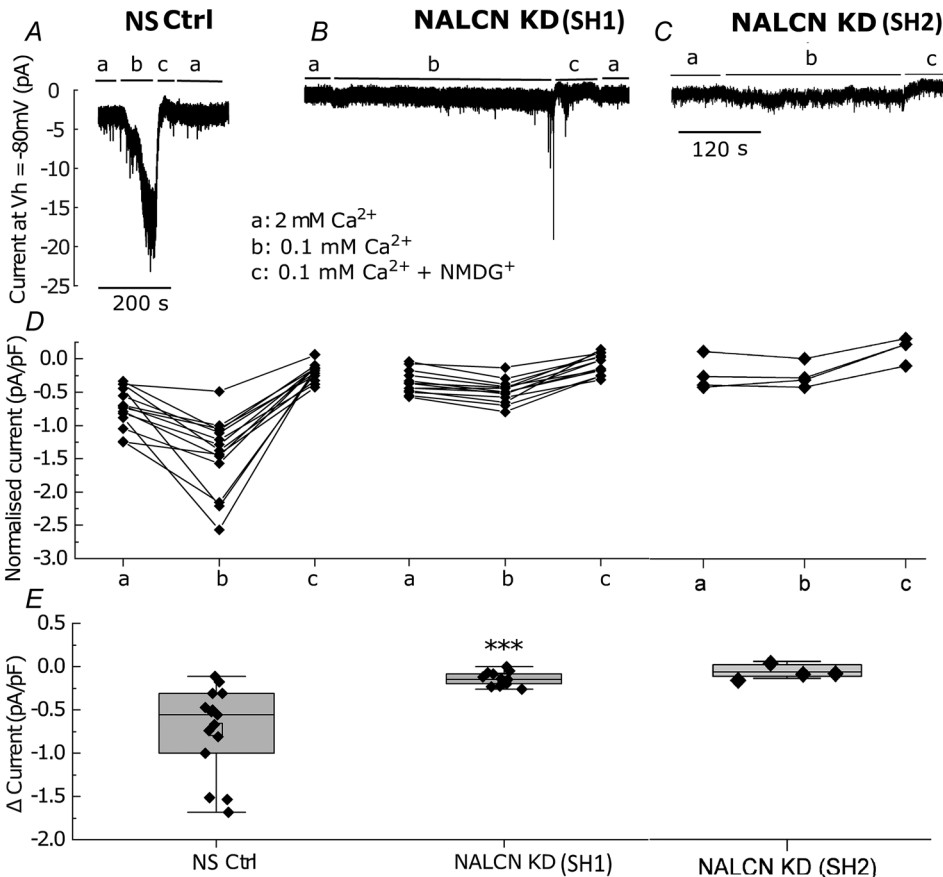

**Figure 6. NALCN is sensitive to changes in extracellular $Ca^{2+}$ level**

*A* and *B*, representative traces of inward leak current in NS Ctrl and NALCN KD pituitary cells and (*C*) in SH2-RNA-mediated NALCN KD cells (SH2-RNA cells) in knockdown validation experiment (see Methods). From baseline conditions (a), reducing $[Ca^{2+}]_e$ level evoked an inward leak current in NS Ctrl but not in NALCN KD pituitary cells and SH2-RNA cells (b). Subsequent replacement of extracellular $Na^+$ with $NMDG^+$ in reduced $[Ca^{2+}]_e$ returned the inward leak current back to near baseline level (c and a). *B*, representative trace of the inward leak current in a NALCN KD pituitary cell; lowering the $[Ca^{2+}]_e$ resulted in a small rise in the inward leak current. Noice that we maintained NALCN KD pituitary cells in low $[Ca^{2+}]_e$ for a longer period of time (>200 s) to rule out any further changes in current during application. Note that the fast transient spikes ($\sim$ −8 to −19 pA) observed between (b) and (c) are electrical artifacts caused by solution switching and does not have physiological relevance. *C*, representative trace of inward leak current in SH2-RNA-mediated NALCN KD cells; as in (*B*), lowering $[Ca^{2+}]_e$ did not evoke an inward leak current (b). Under reduced $[Ca^{2+}]_e$, subsequent replacement of extracellular $Na^+$ with $NMDG^+$ returned the inward leak current to near baseline level (c). *D*, normalised inward leak current in 2 mM $Ca^{2+}$ (a), 0.1 mM $Ca^{2+}$ (b) and 0.1 mM $Ca^{2+}$ + $NMDG^+$ (c) in NS Ctrl (left), NALCN KD SH1 (middle) and SH2-mediated NALCN KD (right) cells. *E*, the overall rise in inward leak current after reducing $[Ca^{2+}]_e$ in NS Ctrl (left) compared to NALCN KD SH1 (middle) and SH2-RNA-mediated NALCN KD (right). ΔMedian ± interquartile in NS Ctrl: −0.6 ± 0.6 pA/pF, *n* = 14 cells; NALCN KD: −0.17 ± 0.1 pA/pF, *n* = 13 cells, from six animals in each condition, *P* = 0.00071, Mann–Whitney test; Δmedian ± interquartile in SH2-RNA-NALCN KD: −0.025 ± 0.08 pA/pF, *n* = 4 cells from three animals. Box: interquartile, Whiskers: range excluding outliers; central line: median, $V_h$ is the holding current at −80 mV.

conductance in anterior endocrine pituitary cells. Our approach revealed that NALCN channel activity is critical for regulating the electrical and $[Ca^{2+}]_i$ activity of these cells.

## NALCN regulates the excitability of mouse anterior pituitary cells

Irrespective of cell-type, spontaneous firing in anterior pituitary cells is crucial to maintain normal physiological functions such as hormone secretion, gene expression and $[Ca^{2+}]_i$ oscillations (Fletcher et al., 2018; Kwiecien & Hammond, 1998; Stojilkovic et al., 2010). However, the molecular identity of the major non-selective cationic channels (NSCC) that critically enables spontaneous firing in these cells remains unresolved (Fletcher et al., 2018). Nevertheless, pharmacological studies revealed that NSCCs are unequivocally $Na^+$-dependent and TTX-resistant/insensitive (Kucka et al., 2010; Liang et al., 2011; Sankaranarayanan & Simasko, 1996; Simasko, 1994; Tomić et al., 2011; Zemkova et al., 2016). Studies using RNA expression profile and non-selective pharmacological manipulations have suggested that TRP channels, as well as HCN channels, may contribute to the NSCC described in pituitary cells and related cell lines (Kučka et al., 2012; Kayano et al., 2019; Kretschmannova et al., 2012; Tomić et al., 2011). However, without specific receptor blockade and/or genetic interventions (e.g. modification of ion channel expression level), these assertions are largely speculative.

A recent study in the GH3 pituitary cell line implicated NALCN as a major contributor to NSCC activity (Impheng et al., 2021). Therefore, we knocked down NALCN activity and used sophisticated electrophysiology and calcium imaging to determine its role in regulating cellular excitability and $[Ca^{2+}]_i$ oscillations in native anterior endocrine pituitary cells. Remarkably, our results revealed that NALCN channel activity determines a major $Na^+$ leak conductance in these cells, and its activity is critical to sustaining spontaneous firing activity. Our results also show that NALCN is the main contributor to the depolarising NSCC conductance in pituitary cells, without which cells remain hyperpolarised-silent. In addition, we show that NALCN is crucial for maintaining spontaneous $[Ca^{2+}]_i$ oscillations in these cells, and is sensitive to $[Ca^{2+}]_e$ level, as previously demonstrated in hippocampal neurons (Lu et al., 2010).

Indeed, we found that NALCN KD pituitary cells are significantly hyperpolarised (by about 15 mV) compared to controls. Moreover, 90% of NALCN KD pituitary cells were completely silent and lost the ability to generate AP or ($[Ca^{2+}]_i$) oscillations. Remarkably, restoring the NALCN-like conductance in NALCN KD cells with dynamic clamp rescued cellular RMP and firing activity to normal levels. By extension, the removal of the dynamic clamp mimic of NALCN conductance immediately returned the RMP to hyperpolarised state, silencing the cells.

Together, these results revealed a critical role for NALCN in anterior endocrine pituitary cells, both for maintaining spontaneous firing and $[Ca^{2+}]_i$ oscillations.

## A small NALCN conductance sustains a large depolarising drive

The comparison between background $Na^+$ inward current in control cells and NALCN KD cells showed that the amount of NALCN conductance knocked down in our experiments was $\sim$0.03 nS. This is consistent with the median of 0.05 nS background $Na^+$ conductance required to reactivate silent NALCN KD cells with dynamic clamp. This dynamic clamp-based estimate does not rely on measurements of small noisy currents (a few pA) in voltage clamp, and therefore provides an estimate that is not affected by poor signal-to-noise ratio (Milescu et al., 2008). The profound effect on electrical activity of such a small conductance is consistent with measurements of high input resistance in pituitary cells, the order of 5 G$\Omega$ (Dubinsky & Oxford, 1984), where a very small variation in the current results in a drastic change in RMP. Indeed, with a reversal potential close to 0 mV for this conductance, at a membrane potential of $-60$ mV, the current through a 0.05 nS NALCN conductance is 3 pA. Therefore, with a membrane resistance of 5 G$\Omega$, the depolarisation as a result of the current is 15 mV, a value that is remarkably consistent with our finding that NALCN KD cells were hyperpolarised by $\sim$15 mV.

We have employed a knockdown approach similar to that used in GH3 cells (Impheng et al., 2021), where an estimated NALCN conductance of $\sim$30 pS/pF was reported. This represents a conductance of about three times larger than our estimate in native pituitary cells (10 pS/pF, given our average pituitary cell capacitance of 5 pF). The discrepancy in conductance may reflect a qualitative and/or quantitative difference in rat GH3 cells ionic/excitability physiology when compared to mouse native pituitary cells. It is also possible that GH3 cells may simply have higher levels of NALCN, or overall ion channel expression compared to native mouse pituitary cells. This would indeed account for the measured discrepancy in baseline conductance (higher in GH3 cells), rendering GH3 cells less sensitive to NALCN knockdown because of their lower membrane resistance: as reflected by the $\sim$5 mV RMP suppression in GH3 cells (Impheng et al., 2021) compared to a $\sim$15 mV hyperpolarisation measured in native cells following NALCN knockdown (in our study).

The discrepancy in conductance may also be a result of our slightly higher extracellular concentration of $Mg^{2+}$ than in Impheng et al., 2021 (1 *vs.* 0.8 mм). Given

that divalent cations can block NALCN (Chua et al., 2020), this may partly explain the differences seen. A limitation of our work is that we have sampled from a heterogeneous population of pituitary cells. Thence, different cell types may express different level of NALCN. If such expression heterogeneity exists, then our estimate of NALCN conductance is an aggregate of the average NALCN conductance expressed by the various cells sampled.

Notably, there is no available estimate for NALCN single-channel conductance in pituitary cells. Therefore, with our knockdown-based approach, we are probably slightly under-estimating NALCN conductance in NALCN KD cells because there may still be some residual channel expression/conductance. Nevertheless, our estimation of 0.05 nS for NALCN conductance implies a low level of active NALCN channel expression in the pituitary cell membrane. This may provide support and a parsimonious explanation as to why in our recordings (Figs 4*A* and 6*A*), as well as in the recording of others (Liang et al., 2011), the non-specific cation current is very noisy. Because NALCN has weak voltage sensitivity (Bouasse et al., 2019; Chua et al., 2020) transient blockade by divalent cations, and possibly random channel gating, would result in relatively large current fluctuations if the number of channels is low.

### NALCN activity is modulated by extracellular Ca$^{2+}$ in pituitary cells

Our results also revealed that NALCN channels in mouse anterior pituitary cells are sensitive to low $[Ca^{2+}]_e$ and mediate a low-$[Ca^{2+}]_e$-induced membrane depolarisation. Consistent with this observation, an increase in $[Ca^{2+}]_e$ caused hyperpolarisation in cultured rat pituitary somatotrophs through mechanisms that do not involve calcium channels (Tsaneva-Atanasova et al., 2007). The $[Ca^{2+}]_e$-evoked membrane excitation that we observed agrees with work in cultured rat lactotrophs and somatotrophs (Sankaranarayanan & Simasko, 1996; Tsaneva-Atanasova et al., 2007), but here we unequivocally revealed that NALCN is directly involved. Our results are also consistent with observations in neurons (Bouasse et al., 2019; Chu et al., 2003; Lu et al., 2010) and work reporting NALCN sensitivity to $[Ca^{2+}]_e$ blockade in GH3 cells (Impheng et al., 2021).

Two different mechanisms have been proposed to underlie the $[Ca^{2+}]_e$ sensing capacity of NALCN. First, this may occur through the Ca$^{2+}$-sensing receptor (CaSR) via a $G_q$-protein-dependent pathway that ultimately induces NALCN phosphorylation by protein kinase C (Lee et al., 2019; Lu et al., 2010). Alternatively, NALCN $[Ca^{2+}]_e$ sensitivity could emerge through direct Ca$^{2+}$ binding within the NALCN pore (Chua et al., 2020). In

our experiments, the inward leak current in the majority of the pituitary cells started to increase within 20 s of $[Ca^{2+}]_e$ removal and continued to increase for the next few minutes until it reached a plateau (Fig. 6*A*). Thus, although the exact mechanism(s) involved in the NALCN sensing of $[Ca^{2+}]_e$ in primary pituitary cells is yet to be determined, the dynamics we observed (with regards to the development of the inward leak current to plateaux level) favours a transduction cascade mechanism over a direct pore effect.

Under both physiological and pathological circumstances, $[Ca^{2+}]_e$ can drop markedly in the brain and blood serum (Ferry et al., 1997; Ren, 2011). This variation in $[Ca^{2+}]_e$ is closely linked with ACTH secretion from pituitary corticotrophs (Fuleihan et al., 1996; Isaac et al., 1984). In addition, physiological range variations in serum Ca$^{2+}$ are associated with drastic changes in ACTH secretion (Fuleihan et al., 1996; Isaac et al., 1984). Our results therefore provide a plausible mechanism for $[Ca^{2+}]_e$ sensing in pituitary cells, and link this with appropriate hormonal secretion under normal and pathological states.

### NALCN modulation by neurohormones: a probable key player in pituitary cell responses to hypothalamic/feedback signals?

Our finding that NALCN activity has a profound effect on pituitary cell excitability raises the possibility that, beyond $[Ca^{2+}]_e$ sensing, NALCN could potentially provide a highly sensitive signalling conduit/target through which hypothalamic neurochemicals/neurohormones could act to regulate hormonal function. In support, NALCN can be regulated by G-protein-dependent and independent signalling pathways (Lu et al., 2009; Philippart & Khaliq, 2018; Swayne et al., 2009), providing a possible intracellular access to its activity by a multitude of receptor systems. Moreover, NALCN is sensitive to hypothalamic neuropeptides, such as substance P and neurotensin (Lu et al., 2009), the activity of which controls hormonal regulation (Eckstein et al., 1980), and, in GH3 pituitary cell lines, NALCN activity promotes basal and TRH-dependent prolactin secretion (Impheng et al., 2021).

These observations are consistent with other *in vitro* work showing that the removal of extracellular Na$^+$ to suppress NSCC activity (which our data now identifies as primarily NALCN-driven) in rat pituitary cells negates the ability of growth hormone-releasing hormone (GHRH) to evoke growth hormone secretion from these cells (Kato et al., 1988), mostly by preventing GHRH-induced Ca$^{2+}$ influx (Lussier et al., 1991; Naumov et al., 1994). This suppression of hormonal secretion is TTX-insensitive and can be rescued by RMP depolarisation with artificially

high extracellular $K^+$ (Kato et al., 1988). This supports the concept of a $Na^+$-dependent depolarising mechanism other than TTX-sensitive voltage-gated $Na^+$ channels for the sensitised hormonal-dependent hormone release by pituitary cells.

Removal of extracellular $Na^+$ in murine corticotroph cells also substantially delayed the stimulatory response of these cells to corticotrophin-releasing hormone (Liang et al., 2011), presumably by suppressing spontaneous $[Ca^{2+}]_i$ oscillations in the cells (Tomić et al., 2011). In addition, progesterone and oestrogen have been shown to regulate NALCN expression in human myometrial cells (Amazu et al., 2020), although this remains to be determined in the pituitary. Therefore, altogether, this raises the possibility that hormones and neuropeptides may modulate pituitary NALCN activity in a multifaceted fashion and timescale to influence hormonal release.

In summary, our targeted NALCN knockdown approach, electrophysiology and calcium imaging methods have highlighted the importance of NALCN activity in supporting appropriate excitability in native pituitary cells. This discovery may provide an important step in our understanding of pituitary cell excitability and its intersection with hormonal regulation. It may also provide a potential target for therapeutic interventions in endocrine-related disorders that are linked with disruptions in pituitary hormone secretion.

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

## Additional information

### Data availability statement

All data generated or analysed in this study are included within the manuscript and the Supporting information.

### Competing interests

The authors declare that they have no competing interests.

### Author contributions

M.B., M.M., A.M., P.G.W., R.P., J.J.W., J.T. and M.D.C.B. conceived and designed the research. MB and MM performed genetic manipulation experiments and M.B. performed electro-physiology experiments as well as calcium imaging and data analysis. M.B., A.M., J.J.W., J.T. and M.D.C.B wrote the paper. All authors have approved the final version of the manuscript submitted for publication and agree to be accountable for all aspects of the work. All persons designated as authors qualify for authorship, and all those who qualify for authorship are listed.

### Funding

This work was supported by a Welcome Trust Institutional Strategic Support Award (WT105618MA) to JT and JJW. MDCB was supported by a grant from the Biotechnology and Biological Sciences Research Council (BBSRC) (BB/S01764X/1 and BB/S01764X/2), and JJW acknowledges financial support from the Medical Research Council (MR/N008936/1). MB was the recipient of a University of Exeter studentship.

### Acknowledgements

We thank members of the University of Exeter Biological Services Unit for their assistance in colony maintenance and husbandry. We also extend our thanks to Professor Stephan Pless and Dr Han Chua, who kindly provided the expression plasmids encoding for NALCN, UNC-79, UNC-80 and FAM155A, as well as Dr Lidiane Perreira Garcia for her kind help with the immunofluorescence experiments shown in Fig. 1*H*.

### Keywords

background sodium channel, electrical activity, endocrine cells, NALCN, non-selective cationic conductance, pituitary cells, sodium leak channels

### Supporting information

Additional supporting information can be found online in the Supporting Information section at the end of the HTML view of the article. Supporting information files available:

**Statistical Summary Document**
**Peer Review History**
**Supporting data**

