## [Peer Review History · The Journal of Physiology]

The background sodium leak channel NALCN is a major controlling factor in pituitary cell excitability

Marziyeh Belal, Mariusz Mucha, Arnaud Monteil, Paul G Winyard, Robert Pawlak, Jamie J Walker, Joel Tabak, and Mino David Belle

DOI: 10.1113/JP284036

Corresponding author(s): Mino Belle (Mino.Belle@manchester.ac.uk)

The following individual(s) involved in review of this submission have agreed to reveal their identity: Axel R. Conception (Referee #1)

Review Timeline:

Submission Date:	05-Jan-2023
Editorial Decision:	08-Feb-2023
Revision Received:	05-May-2023
Editorial Decision:	02-Jun-2023
Revision Received:	23-Jul-2024
Editorial Decision:	06-Aug-2024
Revision Received:	05-Oct-2024
Editorial Decision:	07-Oct-2024
Revision Received:	09-Oct-2024
Accepted:	29-Oct-2024

Senior Editor: Peking Fong

Reviewing Editor: Yamuna Krishnan

Transaction Report:

Dear Dr Belle,

Re: JP-RP-2023-284036 "The background sodium leak channel NALCN is a major controlling factor in pituitary cell excitability" by Marziyeh Belal, Mariusz Mucha, Arnaud Monteil, Paul G Winyard, Robert Pawlak, Jamie J Walker, Joel Tabak, and Mino David Belle

Thank you for submitting your manuscript to The Journal of Physiology. It has been assessed by a Reviewing Editor and by 2 expert referees and we are pleased to tell you that it is acceptable for publication following satisfactory revision.

REVISION CHECKLIST:

We look forward to receiving your revised submission.

Yours sincerely,

Dr Peiyong Fong
Senior Editor
The Journal of Physiology
<https://jp.msubmit.net>
<http://jp.physoc.org>
The Physiological Society
Hodgkin Huxley House
30 Farringdon Lane
London, EC1R 3AW
UK
<http://www.physoc.org>
<http://journals.physoc.org>

REQUIRED ITEMS

- Author photo and profile. First (or joint first) authors are asked to provide a short biography (no more than 100 words for one author or 150 words in total for joint first authors) and a portrait photograph. These should be uploaded and clearly labelled with the revised version of the manuscript. See Information for Authors for further details.
- You must start the Methods section with a paragraph headed Ethical Approval. A detailed explanation of journal policy and regulations on animal experimentation is given in Principles and standards for reporting animal experiments in The Journal of Physiology and Experimental Physiology by David Grundy J Physiol, 593: 2547-2549. doi:10.1113/JP270818). A checklist outlining these requirements and detailing the information that must be provided in the paper can be found at: <https://physoc.onlinelibrary.wiley.com/hub/animal-experiments>. Authors should confirm in their Methods section that their experiments were carried out according to the guidelines laid down by their institution's animal welfare committee, and conform to the principles and regulations as described in the Editorial by Grundy (2015). The Methods section must contain details of the anaesthetic regime: anaesthetic used, dose and route of administration and method of killing the experimental animals.
- The Journal of Physiology funds authors of provisionally accepted papers to use the premium BioRender site to create high resolution schematic figures. Follow this link and enter your details and the manuscript number to create and download figures. Upload these as the figure files for your revised submission. If you choose not to take up this offer we require figures to be of similar quality and resolution. If you are opting out of this service to authors, state this in the Comments section on the Detailed Information page of the submission form. The link provided should only be used for the purposes of this submission. Authors will be charged for figures created on this premium BioRender account if they are not related to this manuscript submission.
- Please upload separate high-quality figure files via the submission form.
- Please ensure that the Article File you upload is a Word file.
- Your paper contains Supporting Information of a type that we no longer publish. Any information essential to an understanding of the paper must be included as part of the main manuscript and figures. The only Supporting Information that we publish are video and audio, 3D structures, program codes and large data files. Your revised paper will be returned to you if it does not adhere to our Supporting Information Guidelines.
- A Statistical Summary Document, summarising the statistics presented in the manuscript, is required upon revision. It must be on the Journal's template, which can be downloaded from the link in the Statistical Summary Document section here: https://jp.msubmit.net/cgi-bin/main.plex?form_type=display_requirements#statistics.
- Papers must comply with the Statistics Policy: https://jp.msubmit.net/cgi-bin/main.plex?form_type=display_requirements#statistics.

In summary:

- If n {less than or equal to} 30, all data points must be plotted in the figure in a way that reveals their range and distribution.

A bar graph with data points overlaid, a box and whisker plot or a violin plot (preferably with data points included) are acceptable formats.

- If $n > 30$, then the entire raw dataset must be made available either as supporting information, or hosted on a not-for-profit repository e.g. FigShare, with access details provided in the manuscript.

- 'n' clearly defined (e.g. x cells from y slices in z animals) in the Methods. Authors should be mindful of pseudoreplication.

- All relevant 'n' values must be clearly stated in the main text, figures and tables, and the Statistical Summary Document (required upon revision).

- The most appropriate summary statistic (e.g. mean or median and standard deviation) must be used. Standard Error of the Mean (SEM) alone is not permitted.

- Exact p values must be stated. Authors must not use 'greater than' or 'less than'. Exact p values must be stated to three significant figures even when 'no statistical significance' is claimed.

- Statistics Summary Document completed appropriately upon revision.

- Please include an Abstract Figure file, as well as the figure legend text within the main article file. The Abstract Figure is a piece of artwork designed to give readers an immediate understanding of the research and should summarise the main conclusions. If possible, the image should be easily 'readable' from left to right or top to bottom. It should show the physiological relevance of the manuscript so readers can assess the importance and content of its findings. Abstract Figures should not merely recapitulate other figures in the manuscript. Please try to keep the diagram as simple as possible and without superfluous information that may distract from the main conclusion(s). Abstract Figures must be provided by authors no later than the revised manuscript stage and should be uploaded as a separate file during online submission labelled as File Type 'Abstract Figure'. Please ensure that you include the figure legend in the main article file. All Abstract Figures should be created using BioRender. Authors should use The Journal's premium BioRender account to export high-resolution images. Details on how to use and access the premium account are included as part of this email.

EDITOR COMMENTS

Reviewing Editor:

A similar manuscript was published recently in FASEB journal using a pituitary gland cell line from rats. The current study by Belal et al. in primary mouse pituitary gland cells confirms previous findings.

Identifying the NALCN as the cation channel responsible for depolarization of the plasma membrane and firing of action potential and subsequent intracellular calcium increase in endocrine anterior pituitary cells. The results provide a plausible mechanism through which hormonal feedback signals from the brain and body could powerfully affect pituitary activity to influence hormonal function.

Senior Editor:

Your manuscript has been reviewed by two Expert Referees and a Reviewing Editor, who appreciate its potential impactfulness toward understanding mouse anterior pituitary cell excitability. Although both referees appreciate this study's objectives, you will also read in their detailed critiques that multiple, major points of concern were identified.

One non-trivial limitation is the study's seemingly confirmatory nature, as qualitatively (but not quantitatively) similar involvement of NALCN was observed previously in GH3 cells, a rat-derived pituitary line. Assumed similarities, as well as subtle distinctions, between the two models deserve greater attention, as noted by Referee 1.

For example, both Referees raise issues concerning the immunofluorescence data presented in Figure 1--both in interpretation and in presenting proper controls. Based on the data shown, the Authors' conclusion that NALCN localizes at the plasma membrane, as noted by Referee 1, is strongly debatable, and should be further probed using the suggested controls. Co-localization with the Na⁺/K⁺ pump is straightforward, as comparative evaluation of staining in control and NALCN knockdown cells. The functional measures, after all, are performed on primary cell cultures rather than intact pituitary.

Knockdown efficiency should be both better controlled and more rigorous (see Referee 1, point 3 and Referee 2, point 1). The experiments suggested are straightforward and would inspire greater confidence in forthcoming conclusions.

Overall, more robust interrogation and controls are recommended to address all concerns raised.

REFeree COMMENTS

Referee #1:

In the study entitled "The background sodium leak channel NALCN is a major controlling factor in pituitary cell excitability", Belal et al. propose that the sodium leak channel NALCN is responsible for the intracellular Ca^{2+} oscillations that are triggered at resting membrane potential (RMP) in primary murine anterior endocrine pituitary cells. By transducing these cells with lentiviruses to target NALCN with shRNAs, the authors found a hyperpolarization of their RMP which suppresses their firing and intracellular Ca^{2+} oscillations. The authors then concluded that NALCN is responsible for controlling pituitary cell excitability. The endocrine pituitary cells have been known to be electrically excitable for over 45 years but the set of ion channels that control this process remains unknown. A recent publication using a rat-derived pituitary endocrine cell line GH3 suggested that NALCN was responsible for their cellular excitability (Impheng et al, 2021). The study by Belal confirms that NALCN is potentially the sodium leak channel responsible for cell excitability in primary pituitary cells in mice as observed in the rat cell line from the previous study. The manuscript is very interesting and relevant to the understanding of the mechanisms of cellular excitability in pituitary gland cells. My main concern is that both studies have used the same approach (shRNA knockdown of NALCN) with the same shRNA-encoded lentiviral plasmid. In my opinion, I consider that other alternative approaches (CRISPR deletion or the use of KO cells derived from mice with genetic deletion of NALCN) will be necessary to validate these findings in future studies. In order to enhance the scope of the manuscript, please consider addressing the following major points:

1. Although the authors mentioned that Gd^{3+} suppresses spontaneous firing in pituitary cells (lines 380-381) [the authors are referring to studies using cell lines obtained from rat], I will suggest to validate these findings in the mouse primary pituitary cells.
2. In Figure 1, the authors show immunofluorescent staining of NALCN using IHC. A negative control for antibody staining is shown in panel H in which the primary antibody was omitted for the staining. First, the staining with this antibody seems to localize in the cytosol rather than the cell surface, where the staining of NALCN should be expected. Second, the negative control shown in panel H is not a proper control. I will recommend the authors to: i) use a cell membrane specific antibody (e.g. $\text{Na}^+\text{K}^+\text{-ATPase}$) as a control for surface expression and ii) use shRNA-mediated knockdown of NALCN to validate antibody staining (rather than showing a staining that lacks primary antibody). In this regard, the authors will have to use a secondary antibody conjugated with another fluorochrome different (other than green) since the lentivirus used to deliver shRNA expresses eGFP (as indicated in line 396). This will be a proper control to validate antibody staining of NALCN in the murine pituitary gland cells but also will validate the knockdown of NALCN with the shRNAs.
3. As mentioned in point 2, the authors have not shown the data of NALCN knockdown but rather a lack of functionality (hyperpolarization of the cells and lack of Ca^{2+} oscillations). These results don't rule out the possibility of a potential off-target effect by shRNAs. As mentioned before, the authors used one single shRNA that targets rat NALCN, whose sequence is conserved in mouse, but a BLAST analysis of that target sequence (GCAACAGACTGTGGCAATT) shows a bunch of potential off-targets in mouse transcripts. It would be important, in my opinion, to validate part of these phenotypic findings with another shRNA targeting a different sequence of mouse NALCN to rule-out the possibility of an off-target effect by shRNA knockdown system. Also, did the authors observe any differences in the survival or appearance of the cells upon shRNA-mediated knockdown of NALCN? The authors indicate they have used a scramble shRNA as a control throughout the text. This annotation is incorrect since the authors indicate they have used a non-silencing control shRNA (Dharmacon #RHS4346). Technically, a scramble shRNA will have exact same nucleotides of the target shRNA but scrambled, and this is not the case in this study. Therefore I will suggest the authors to use control shRNA rather than scramble. Usually this non-silencing control targets non-mammalian genes, and they are fine to use.
4. In Figure 2D, the authors show statistical significance (***) for the % of active cells in the NALCN KD group. What statistical test was used to calculate these differences if these values correspond to one data point. Was this experiment repeated multiple times? If so, please show data point distribution. If not, please remove the stars.
5. The authors show that NALCN KD cells display hyperpolarization of their RMP by ~ 20 mV, which indeed is a very interesting finding. In the previous study by Impheng et al, 2021, the authors found a slight hyperpolarization of their RMP by ~ 5 mV. I know that in this study the authors used mouse primary cells while the previous study by Impheng et al, 2021 rat cell lines were used. Can the authors speculate in the discussion section about these differences in the lines 759-771 where they address the differences between both studies? Interestingly, the stronger hyperpolarization in the case of the NALCN-KD in mouse cells depletes the firing frequency while a slight hyperpolarization, in the case of the NALCN-KD rat cell line, there is just a reduction in the frequency but not complete suppression.
6. In Figure 5A,B, the authors used 15 mM extracellular K^+ solution to depolarize the cells causing a transient increase in cytosolic Ca^{2+} concentration. I am wondering how strong is such depolarization. What is the theoretical V_m achieved in this condition? Can the authors measure the V_m upon exposing the cells to 15 mM $[\text{K}^+]_o$? Please add arbitrary units to the y-axes in $[\text{Ca}^{2+}]_i$ in panel 5A, or just change to F340/380.

Minor

1. End of line 210, please add "performed" between were as.

2. Please indicate the abbreviations only once when first used (e.g. RMP) Please replace depolarisation by depolarization, and hyperpolarisation by hyperpolarization.

3. Please remove the description details of experiments in lines 403-406, 464-466, 519-529, and 600-601 from the main text since they are redundant with the Figure Legend 2, 3, 4 and 5, respectively. Keep info in one place but not in both.

Referee #2:

This study aims to determine ion channels that regulate resting membrane potential (RMP) to close to the firing threshold of action potentials to regulate intracellular calcium ($[Ca^{2+}]_i$) level, an essential signaling conduit for hormonal secretion in endocrine anterior pituitary cells. Using viral transduction knocking down NALCN combined with electrophysiology methods and calcium imaging, the results showed that NALCN (Sodium Leak Channel, Non-Selective) forms the major Na^+ leak conductance in these cells, appropriately tuning cellular RMP for sustaining spontaneous firing activity and therefore controlling pituitary cell excitability. The results provide a plausible mechanism through which hormonal feedback signals from the brain and body could powerfully affect pituitary activity to influence hormonal function. The results obtained with sound methodology (electrophysiological approaches) support the conclusions and claims. Some minor suggestions:

1. Related to Figure 2, the efficiency of Knocking down NALCN should be checked by western blot.
2. In Figure 1, a negative control should be shown parallel to A and indicates where the H is taken.
3. On line 379, the subtitle is not necessary and the content under this subtitle (lines 380-392) should be combined with the content under the subtitle on line 394.
4. The subtitle of Figure 3 A) should be like " A representative voltage trace of silent NALCN KD pituitary cells with firing activity being restored" because it is not just a representative voltage trace of silent NALCN KD pituitary cells.
5. For the y-axis label in Figure 5 A, it should be indicated it is a ratio of 340/380 before $[Ca^{2+}]_i$ or concentration unit after $[Ca^{2+}]_i$.
6. On line 646 - 647, "low" should be added before $[Ca^{2+}]_e$ -induced depolarization

END OF COMMENTS

Confidential Review

05-Jan-2023

In the study entitled “The background sodium leak channel NALCN is a major controlling factor in pituitary cell excitability”, Belal et al. propose that the sodium leak channel NALCN is responsible for the intracellular Ca^{2+} oscillations that are triggered at resting membrane potential (RMP) in primary murine anterior endocrine pituitary cells. By transducing these cells with lentiviruses to target NALCN with shRNAs, the authors found a hyperpolarization of their RMP which suppresses their firing and intracellular Ca^{2+} oscillations. The authors then concluded that NALCN is responsible for controlling pituitary cell excitability. The endocrine pituitary cells have been known to be electrically excitable for over 45 years but the set of ion channels that control this process remains unknown. A recent publication using a rat-derived pituitary endocrine cell line GH3 suggested that NALCN was responsible for their cellular excitability (Impheng et al, 2021). The study by Belal confirms that NALCN is potentially the sodium leak channel responsible for cell excitability in primary pituitary cells in mice as observed in the rat cell line from the previous study. The manuscript is very interesting and relevant to the understanding of the mechanisms of cellular excitability in pituitary gland cells. My main concern is that both studies have used the same approach (shRNA knockdown of NALCN) with the same shRNA-encoded lentiviral plasmid. In my opinion, I consider that other alternative approaches (CRISPR deletion or the use of KO cells derived from mice with genetic deletion of NALCN) will be necessary to validate these findings in future studies. In order to enhance the scope of the manuscript, please consider addressing the following major points:

1. Although the authors mentioned that Gd^{3+} suppresses spontaneous firing in pituitary cells
(lines 380-381) [the authors are referring to studies using cell lines obtained from rat], I will suggest to validate these findings in the mouse primary pituitary cells.
2. In Figure 1, the authors show immunofluorescent staining of NALCN using IHC. A negative control for antibody staining is shown in panel H in which the primary antibody was omitted for the staining. First, the staining with this antibody seems to localize in the cytosol rather than the cell surface, where the staining of NALCN should be expected. Second, the negative control shown in panel H is not a proper control. I will recommend the authors to: i) use a cell membrane specific antibody (e.g. $Na^{+}K^{+}$ -ATPase) as a control for surface expression and ii) use shRNA-mediated knockdown of NALCN to validate antibody staining (rather than showing a staining that lacks primary antibody). In this regard, the authors will have to use a secondary antibody conjugated with another fluorochrome different (other than green) since the lentivirus used to deliver shRNA expresses eGFP (as indicated in line 396). This will be a proper control to validate

antibody staining of NALCN in the murine pituitary gland cells but also will validate the knockdown of NALCN with the shRNAs.

3. As mentioned in point 2, the authors have not shown the data of NALCN knockdown but rather a lack of functionality (hyperpolarization of the cells and lack of Ca²⁺ oscillations). These results don't rule out the possibility of a potential off-target effect by shRNAs. As mentioned before, the authors used one single shRNA that targets rat NALCN, whose sequence is conserved in mouse, but a BLAST analysis of that target sequence (GCAACAGACTGTGGCAATT) shows a bunch of potential off-targets in mouse transcripts. It would be important, in my opinion, to validate part of these phenotypic findings with another shRNA targeting a different sequence of mouse NALCN to rule-out the possibility of an off-target effect by shRNA knockdown system. Also, do the authors observed any differences in the survival or appearance of the cells upon shRNA-mediated knockdown of NALCN? The authors indicate they have used a scramble shRNA as a control throughout the text. This annotation is incorrect since the authors indicate they have used a non-silencing control shRNA (Dharmacon #RHS4346). Technically, a scramble shRNA will have exact same nucleotides of the target shRNA but scrambled, and this is not the case in this study. Therefore I will suggest the authors to use control shRNA rather than scramble. Usually this non-silencing controls target non mammalian genes, and they are fine to use.
4. In Figure 2D, the authors show statistical significance (***) for the % of active cells in the NALCN KD group. What statistical test was used to calculate these differences if these values correspond to one data point. Was this experiment repeated multiple times? If so, please show data point distribution. If not, please remove the stars.
5. The authors show that NALCN KD cell display hyperpolarization of their RMP by ~-20 mV, which indeed is a very interesting. In the previous study by Impheng et al, 2021, the authors found a slight hyperpolarization of their RMP by ~-5 mV. I know that in this study the authors used mouse primary cells while the previous study by Impheng et al, 2021 rat cell lines were used. Can the authors speculate in the discussion section about these differences in the lines 759-771 where they address the differences between both studies? Interestingly, seems like stronger hyperpolarization in the case of the NALCN-KD in mouse cells depletes the firing frequency while a slight hyperpolarization, in the case of the NALCN-KD rat cell line, there is just a reduction in the frequency but not complete suppression.

6. In Figure 5A,B, the authors used 15 mM extracellular K⁺ solution to depolarize the cells causing a transient increase in cytosolic Ca²⁺ concentration. I am wondering how strong is such depolarization. What is the theoretical V_m achieved in this condition? Can the authors measure the V_m upon exposing the cells to 15 mM [K⁺]_o? Please add arbitrary units to the y-axes in [Ca²⁺]_i in panel 5A, or just change to F340/380

Minor

1. End of line 210, please add “performed’ between were as.
2. Please indicate the abbreviations only once when first used (e.g. RMP) Please replace depolarisation by depolarization, and hyperpolarisation by hyperpolarization.
3. Please remove the description details of experiments in lines 403-406, 464-466, 519-529, and 600-601 from the main text since they are redundant with the Figure Legend 2, 3, 4 and 5, respectively. Keep info in one place both not in both.

Review on “*The background sodium leak channel NALCN is a major controlling factor in pituitary cell excitability*” by Marziyeh Belal, Mariusz Mucha, Arnaud Monteil, Paul G Winyard, Robert Pawlak, Jamie J Walker, Joel Tabak, and Mino David Belle (JP-RP-2023-284036)

Remarks to the author.

This study aims to determine ion channels that regulate resting membrane potential (RMP) to close to the firing threshold of action potentials to regulate intracellular calcium ($[Ca^{2+}]_i$) level, an essential signaling conduit for hormonal secretion in endocrine anterior pituitary cells. Using viral transduction knocking down NALCN combined with electrophysiology methods and calcium imaging, the results showed that NALCN (Sodium Leak Channel, Non-Selective) forms the major Na^+ leak conductance in these cells, appropriately tuning cellular RMP for sustaining spontaneous firing activity and therefore controlling pituitary cell excitability. The results provide a plausible mechanism through which hormonal feedback signals from the brain and body could powerfully affect pituitary activity to influence hormonal function. The results obtained with sound methodology (electrophysiological approaches) support the conclusions and claims. Some minor suggestions:

1. Related to figure 2, the efficiency of Knocking down NALCN should be checked by western blot.
2. In Figure 1, a negative control should be shown parallel to **A** and indicates where the **H** is taken.
3. On line 379, the subtitle is not necessary and the content under this subtitle (lines 380-392) should be combined with the content under the subtitle on line 394.
4. The subtitle of Figure 3 A) should be like “ A representative voltage trace of silent NALCN KD pituitary cells with firing activity being restored” because it is not just a representative voltage trace of silent NALCN KD pituitary cells.
5. For the y-axis label in Figure 5 A, it should be indicated it is a ratio of 340/380 before $[Ca^{2+}]_i$ or concentration unit after $[Ca^{2+}]_i$.
6. On line 646 - 647, “low” should be added before $[Ca^{2+}]_e$ -induced depolarization

Remarks to the editor.

The paper is technically sound. The claims are appropriately discussed in the context of the previous literature. The manuscript was clearly written and there is no special ethical concern arising from the use of animals (mice).

Suggestion: accepted with minor revision.

Comments on Influence

Quite influential - identifying the NALCN as the cation channel responsible for depolarization of the plasma membrane and firing of action potential and subsequent intracellular calcium increase in endocrine anterior pituitary cells. The results provide a plausible mechanism through which hormonal feedback signals from the brain and body could powerfully affect pituitary activity to influence hormonal function.

Dear Editors,

We would like to thank you for your continuing interest in our manuscript, and we are delighted to learn that it is acceptable for publication in the Journal of Physiology and by the positive comments received.

We also sincerely thank the reviewers for their astute and constructive comments and suggestions, and their time. We have addressed them as detailed below, and accordingly revised our manuscript.

We are also providing additional data where appropriate in this response letter to address the reviewer's comments. We feel best to provide these here, but will be equally happy to incorporate relevant figures in the manuscript should the Editors and/or reviewers wish.

As requested, we are providing two versions of the manuscript. One with all relevant changes made to the document highlighted with blue text, and the second is a clean version with no changes tracked.

We trust that our revisions sufficiently addressed the reviewer's comments.

With best wishes,

Mino Belle

EDITOR COMMENTS

Reviewing Editor:

A similar manuscript was published recently in FASEB journal using a pituitary gland cell line from rats. The current study by Belal et al. in primary mouse pituitary gland cells confirms previous findings.

Identifying the NALCN as the cation channel responsible for depolarization of the plasma membrane and firing of action potential and subsequent intracellular calcium increase in endocrine anterior pituitary cells. The results provide a plausible mechanism through which hormonal feedback signals from the brain and body could powerfully affect pituitary activity to influence hormonal function.

Senior Editor:

Your manuscript has been reviewed by two Expert Referees and a Reviewing Editor, who appreciate its potential impactfulness toward understanding mouse anterior pituitary cell excitability. Although both referees appreciate this study's objectives, you will also read in their detailed critiques that multiple, major points of concern were identified.

One non-trivial limitation is the study's seemingly confirmatory nature, as qualitatively (but not quantitatively) similar involvement of NALCN was observed previously in GH3 cells, a rat-derived pituitary line. Assumed similarities, as well as subtle distinctions, between the two models deserve greater attention, as noted by Referee 1.

For example, both Referees raise issues concerning the immunofluorescence data presented in

Figure 1--both in interpretation and in presenting proper controls. Based on the data shown, the Authors' conclusion that NALCN localizes at the plasma membrane, as noted by Referee 1, is strongly debatable, and should be further probed using the suggested controls. Co-localization with the Na⁺/K⁺ pump is straightforward, as comparative evaluation of staining in control and NALCN knockdown cells. The functional measures, after all, are performed on primary cell cultures rather than intact pituitary.

Knockdown efficiency should be both better controlled and more rigorous (see Referee 1, point 3 and Referee 2, point 1). The experiments suggested are straightforward and would inspire greater confidence in forthcoming conclusions.

Overall, more robust interrogation and controls are recommended to address all concerns raised.

REFeree COMMENTS

Referee #1:

In the study entitled "The background sodium leak channel NALCN is a major controlling factor in pituitary cell excitability", Belal et al. propose that the sodium leak channel NALCN is responsible for the intracellular Ca²⁺ oscillations that are triggered at resting membrane potential (RMP) in primary murine anterior endocrine pituitary cells. By transducing these cells with lentiviruses to target NALCN with shRNAs, the authors found a hyperpolarization of their RMP which suppresses their firing and intracellular Ca²⁺ oscillations. The authors then concluded that NALCN is responsible for controlling pituitary cell excitability. The endocrine pituitary cells have been known to be electrically excitable for over 45 years but the set of ion channels that control this process remains unknown. A recent publication using a rat-derived pituitary endocrine cell line GH3 suggested that NALCN was responsible for their cellular excitability (Impheng et al, 2021). The study by Belal confirms that NALCN is potentially the sodium leak channel responsible for cell excitability in primary pituitary cells in mice as observed in the rat cell line from the previous study. The manuscript is very interesting and relevant to the understand the mechanisms of cellular excitability in pituitary gland cells. My main concern is that both studies have used the same approach (shRNA knockdown of NALCN) with the same shRNA-encoded lentiviral plasmid. In my opinion, I consider that other alternative approaches (CRISPR deletion or the use of KO cells derived from mice with genetic deletion of NALCN) will be necessary to validate these findings in future studies. In order to enhance the scope of the manuscript, please consider addressing the following major points:

We thank the reviewer for finding our manuscript interesting and relevant to understanding the mechanisms of pituitary gland cellular excitability. We concur that the use of an alternative approach, such as the use of CRISPR in future studies, for NALCN genetic deletion will be beneficial.

1. Although the authors mentioned that Gd³⁺ suppresses spontaneous firing in pituitary cells (lines 380-381) [the authors are referring to studies using cell lines obtained from rat], I will suggest to validate these finding in the mouse primary pituitary cells.

We thank the reviewer for highlighting this and we have removed the study with Gd3+ here. We wanted to draw attention that several studies have shown the importance of non-selective cationic conductance in pituitary cell's excitability, but the identity of the channels remained unknown. As such we have appropriately altered the text to reflect a broader statement, including studies supporting the activity of non-selective cationic conductance in pituitary cells as part of our opening motivation for studying NALCN. The opening paragraph now reads "*Several studies have shown that functional non-selective cationic conductances (NSCC) are responsible for excitability in pituitary cells (Stojilkovic et al, 2010; Fletcher et al, 2018), but the identity of the channels responsible remains elusive.*", (Page 10, lines 383 to 385).

2. In Figure 1, the authors show immunofluorescent staining of NALCN using IHC. A negative control for antibody staining is shown in panel H in which the primary antibody was omitted for the staining. First, the staining with this antibody seems to localize in the cytosol rather than the cell surface, where the staining of NALCN should be expected.

We thank the reviewer for such astute comments, and we have segregated our response below for clarity and convenience.

We believe that it is common for 4-domain ion channels such as NALCN to be localised in the intracellular compartments, as well as on the cell surface, where they are associated with intracellular structures, such as the endoplasmic reticulum and Golgi apparatus, serving numerous intracellular functions including processes linked with the regulation of the ion channel's own synthesis and modification (Zhang et al, 2012; Montell, 2006; also reviewed by Dong et al, 2010). As such, their intracellular distribution is not surprising. We believe that our observation of cellular localisation of NALCN-immunofluorescence is consistent with previous work, example (Zhang et al., 2021; Li et al 2021) which show similar cellular immunofluorescence distribution. Based on the reviewer's comment, we have now referenced these authors in our manuscript (Page 9, line 346 - 347). Our intension was to simply begin our investigation by showing that the NALCN protein is present in pituitary cells with the intension to use sensitive electrophysiology methods to provide evidence of their functional characteristics.

In addition, one of the co-authors, Arnaud Monteil, has performed NALCN immunofluorescence in HEK293T cells following transfection of NALCN-GFP along with its subunits (NALCN+UNC-79+UNC-80+FAM155A), which we are providing as two separate experimental replicates in the below figures (Figure1A and B below). Although we appreciate that this is in HEK293T cells, please note a large fraction of NALCN is intracellular, which as mentioned above, from our point of view is consistent with work by (e.g. Zhang et al; Li et al., 2021). Please see our response to antibody specificity below.

Figure 1

Second, the negative control shown in panel H is not a proper control. I will recommend the authors to: i) use a cell membrane specific antibody (e.g. Na⁺K⁺-ATPase) as a control for surface expression and ii) use shRNA-mediated knockdown of NALCN to validate antibody staining (rather than showing a staining that lack primary antibody). In this regard, the authors will have to use a secondary antibody conjugated with another fluorochrome different (other than green) since the lentivirus used to deliver shRNA expresses eGFP (as indicated in line 396). This will be a proper control to validate antibody staining of NALCN in the murine pituitary gland cells but also will validate the knockdown of NALCN with the shRNAs.

We thank the reviewer for raising this point, and we are happy to report that the specificity of this antibody against NALCN used in our study has previously been validated by previous studies, both through pre-absorption and in conditional knockout mice for NALCN (Zhang et al., 2021; Li et al., 2021). We apologise for not providing information regarding this in our manuscript and as such, we have added the following to our results section “*that is specific to the NALCN protein (Zhang et al., 2021; Li et al., 2021)*”, (Page 9, lines 346 - 347). Please see our above response regarding cellular localisation.

In addition, we also provide the above figure (Figure 1) from two experimental replicates (please see figure 1A and B above) to show NALCN immunofluorescence (NALCN-IF) in HEK293T cells following transfection of NALCN-GFP along with its subunits (NALCN+UNC-79+UNC-80+FAM155A) (provided by our co-author Arnaud Monteil). We observed 1:1 colocalization of signals (NALCN-IF and NALCN-GFP). As the reviewer pointed out, here we used a different colour fluorophore to localise NALCN-primary antibody reaction as GFP reports successful NALCN transfection (Figure 1 above).

With regards to antibody validation with immunofluorescence staining in cells following NALCN knock-down: Our electrophysiology recordings (dynamic clamp, current and voltage clamp) strongly suggest that in the primary pituitary cells only a small fraction of NALCN channels needs to be absent (knocked down) to produce such profound suppressive electrical effects on the cells. We therefore do not believe that immunofluorescence will be sensitive enough to detect such changes in ion channel fraction. As mentioned above, we have referenced previous work where the antibody we used is validated in conditional knockout mice (Zhang et al., 2021; Li et al., 2021).

We agree that panel H is not showing the appropriate control and have now removed this panel from figure 1.

3. As mentioned in point 2, the authors have not shown the data of NALCN knockdown but rather a lack of functionality (hyperpolarization of the cells and lack of Ca²⁺ oscillations). These results don't rule out the possibility of a potential off-target effect by shRNAs. As mentioned before, the authors used one single shRNA that targets rat NALCN, whose sequence is conserved in mouse, but a BLAST analysis of that target sequence (GCAACAGACTGTGGCAATT) shows a bunch of potential off-targets in mouse transcripts. It would be important, in my opinion, to validate part of these phenotypic findings with another shRNA targeting a different sequence of mouse NALCN to rule-out the possibility of an off-target effect by shRNA knockdown system.

We thank the reviewer for this comment and appreciate the concern. As such we are providing data below where we have used another shRNA (SH2-RNA or sh/miR-30) that targets a different sequence of mouse NALCN (5' - TTAATCCAGAGTATGTCAG - 3' (Dharmacon # V2LMM_77139). Indeed, as with our shRNA results, SH2-RNA-mediated NALCN KD hyperpolarised and silenced the pituitary cells when compared with controls (please see figure 2 below). This SH2-RNA-mediated activity suppression is consistent with the results we obtained with the shRNA knock-down (please see figures 2 and 3 and associated text in the manuscript). In addition, diminishing NALCN activity by SH2-RNA rendered the cells insensitive to reduced extracellular calcium, and response that is known to be NALCN-activity dependent and lacking if the activity of NALCN is suppressed (please see figure 6 and associate text). This is consistent with our observation with shRNA-mediated knock-down.

Figure 2

Figure 2: Targeted Patch-clamp recordings of mouse pituitary cells **A)** Resting membrane potential of untreated control cells, and cells where NALCN channels are knock-down (NALCN KD) with SH2-RNA (data are presented as mean \pm SD, untreated control: -45.20 ± 8.2 mV, $n=20$ cells vs SH2-RNA Ctrl: -63.43 mV \pm 2.7 mV, $n=7$ cells, from 8 animals in control condition and 3 animals in SH2RNA condition, $p < 0.0001$; t test – transfected cells identified by GFP). **B)** A representative voltage trace of silent SH2-RNA-mediated NALCN KD pituitary cells, with firing activity being restored once the

nonselective cationic conductance (g) was injected in the cells (note the small amount of conductance, 0.04 nS, needed to restore firing in this cells, identified by the grey box). C) The distribution of added conductance (g) values required for restoring firing activity in hyperpolarised-silent SH2-RNA NALCN KD pituitary cells. Median: 0.05 nS, n=7 cells from 3 animals. D) A representative trace of inward leak current in SH2-RNA-mediated NALCN KD cells. From baseline conditions (a), reducing extracellular Ca^{2+} level did not evoke an inward leak current in SH2-RNA-mediated NALCN KD pituitary cells (b). Under reduced extracellular Ca^{2+} , subsequent replacement of extracellular Na^+ with NMDG⁺ returned the inward leak current to near baseline level (c). E) Summary of average inward leak current (measured at $V_h = -80$ mV) in 2mM Ca^{2+} (condition a), 0.1 mM Ca^{2+} (condition b), and 0.1 mM Ca^{2+} + NMDG⁺ (condition c) in SH2-RNA-mediated NALCN KD pituitary cells.

Also, do the authors observed any differences in the survival or appearance of the cells upon shRNA-mediated knockdown of NALCN?

We thank the reviewer for this interesting question. On appearance the cells looked normal and did not seem to have any survival issues when compared with non-silencing controls. However, we did not perform a systematic investigation of this to inform a robust conclusion as it was outside the scope of our investigation.

The authors indicate they have used a scramble shRNA as a control throughout the text. This annotation is incorrect since the authors indicate they have used a non-silencing control shRNA (Dharmacon #RHS4346). Technically, a scramble shRNA will have exact same nucleotides of the target shRNA but scrambled, and this is not the case in this study. Therefore, I will suggest the authors to use control shRNA rather than scramble. Usually, this non-silencing controls target non mammalian genes, and they are fine to use.

We completely agree here with the reviewer and thank you for this suggestion. We have now changed the term scrambled to non-silencing control (NSC) throughout the text and figures as suggested.

4. In Figure 2D, the authors show statistical significance (***) for the % of active cells in the NALCN KD group. What statistical test was used to calculate these differences if these values correspond to one data point. Was this experiment repeated multiple times? If so, please show data point distribution. If not, please remove the stars.

We thank the reviewer for spotting this mistake. No stats were performed here, and we have removed the asterisks. We are just reporting the percentages of active cells under the three conditions.

5. The authors show that NALCN KD cell display hyperpolarization of their RMP by ~ -20 mV, which indeed is a very interesting. In the previous study by Impheng et al, 2021, the authors found a slight hyperpolarization of their RMP by ~ -5 mV. I know that in this study the authors used mouse primary cells while the previous study by Impheng et al, 2021 rat cell lines were used. Can the authors speculate in the discussion section about these differences in the lines 759-771 where they address the differences between both studies?

Interestingly, seems like stronger hyperpolarization in the case of the NALCN-KD in mouse cells depletes the firing frequency while a slight hyperpolarization, in the case of the NALCN-KD rat cell line, there is just a reduction in the frequency but not complete suppression.

We thank the reviewer for such an astute comment and observation here, and we agree that the reader would benefit with a reference to this in the discussion, which we have now added. We agree that this is remarkable and speculate that this may be associated with a qualitative and/or quantitative difference in the ionic physiology of these two cell systems. We have therefore speculated in the discussion possible underpinning mechanisms involved (Page 21 lines 681-690).

Incidentally, we have a separate manuscript under consideration where we focus our investigation on the GH4 cell line with dynamic clamp electrophysiology. We have indeed noticed some quantitative differences, in that cells from the GH4 line required on average double the subtraction of depolarising conductance (0.04 nS) compared with primary pituitary cells (0.02 nS) to induce cell silencing. We speculated that GH4 cells may have a higher expression of ion channels given their hypertriploid property (60+ chromosomes) and presumably may also have lower membrane resistance, therefore require larger depolarising/hyperpolarising forces to alter their excitability. Of note, we also found higher spontaneous firing rates in GH4 cells when compared with activity in the primary pituitary cells, which seems to support the quantitative difference in the ionic physiology of these two cell systems.

6. In Figure 5A,B, the authors used 15 mM extracellular K⁺ solution to depolarize the cells causing a transient increase in cytosolic Ca²⁺ concentration. I am wondering how strong is such depolarization. What is the theoretical V_m achieved in this condition?

Using the Nernst equation with appropriate parameter values: e.g., Temperature: 293 Kelvin, Z:1, [K⁺]_{in}: 150 mM, [K⁺]_{out}: 15 mM the theoretical value calculated is -58 mV which is within the membrane potential range for firing (please see e.g., Figure 2 A and B), as illustrated by the rise in intracellular calcium (Figure 5B).

Can the authors measure the V_m upon exposing the cells to 15 mM [K⁺]_o?

Our aim was to use the depolarising effect of high extracellular K⁺ as a positive control to simply assess the viability of the cells that has become silent by NALCN knock-down, since we did not have a direct electrophysiology readout from these cells. This concentration of extracellular K⁺ (15 mM) is routinely used in electrophysiology experiment for similar reasons. As we mentioned above, theoretically this should depolarise pituitary cells close to -58 mV. Sadly, due to change in circumstances we can no longer perform this specific experiment in a timely manner.

Please add arbitrary units to the y-axis in [Ca²⁺]_i in panel 5A, or just change to F340/380.

We thank the reviewer for noticing this mistake. Since reviewer 2 also raised this point we have now changed this axis label to F340/380 [Ca²⁺]_i ratio.

Minor

1. End of line 210, please add "performed" between were as.

We thank the reviewer for identifying this which we have now corrected (Page 6, line 215).

2. Please indicate the abbreviations only once when first used (e.g. RMP) Please replace depolarisation by depolarization, and hyperpolarisation by hyperpolarization.

We apologize for such mistakes and have corrected these throughout the manuscript.

3. Please remove the description details of experiments in lines 403-406, 464-466, 519-529, and 600-601 from the main text since they are redundant with the Figure Legend 2, 3, 4 and 5, respectively. Keep info in one place but not in both.

We thank the reviewer and we have removed any duplication at these locations.

References:

Dong, X.-et al., (2010) "TRP channels of intracellular membranes," *Journal of Neurochemistry*, 113(2), pp. 313–328. Available at: <https://doi.org/10.1111/j.1471-4159.2010.06626.x>.

Fletcher, P.A. et al., (2018). Common and diverse elements of ion channels and receptors underlying electrical activity in endocrine pituitary cells. *Mol. Cell. Endocrinol.* 463, 23–36. <https://doi.org/10.1016/j.mce.2017.06.022>

Li, J. et al. (2021) "Elevated expression and activity of sodium leak channel contributes to neuronal sensitization of inflammatory pain in rats," *Frontiers in Molecular Neuroscience*, 14. Available at: <https://doi.org/10.3389/fnmol.2021.723395>.

Montell, C. (2006) "An exciting release on TRPM7," *Neuron*, 52(3), pp. 395–397. Available at: <https://doi.org/10.1016/j.neuron.2006.10.012>.

Stojilkovic, S.S. et al., (2010). Ion Channels and Signaling in the Pituitary Gland. *Endocr. Rev.* 31, 845–915. <https://doi.org/10.1210/er.2010-0005>

Zhang, D. et al. (2021) "Sodium leak channel contributes to neuronal sensitization in neuropathic pain," *Progress in Neurobiology*, 202, p. 102041. Available at: <https://doi.org/10.1016/j.pneurobio.2021.102041>.

Zhang, X., Li, X. and Xu, H. (2012) "Phosphoinositide isoforms determine compartment-specific ion channel activity," *Proceedings of the National Academy of Sciences*, 109(28), pp. 11384–11389. Available at: <https://doi.org/10.1073/pnas.1202194109>.

Referee #2:

This study aims to determine ion channels that regulate resting membrane potential (RMP) to close to the firing threshold of action potentials to regulate intracellular calcium ([Ca²⁺]_i) level, an essential signaling conduit for hormonal secretion in endocrine anterior pituitary cells. Using viral transduction knocking down NALCN combined with electrophysiology methods and calcium imaging, the results showed that NALCN (Sodium Leak Channel, Non-Selective) forms the major Na⁺ leak conductance in these cells, appropriately tuning cellular RMP for sustaining spontaneous firing activity and therefore controlling pituitary cell excitability. The results provide a plausible mechanism through which hormonal feedback signals from the brain and body could powerfully affect pituitary activity to influence hormonal function. The results obtained with sound methodology (electrophysiological approaches) support the conclusions and claims. **Some minor suggestions:**

We thank the reviewer for seeing our manuscript having a sound methodology and results that support our main conclusions and claims.

Please find below our responses to your minor comments:

1. Related to Figure 2, the efficiency of Knocking down NALCN should be checked by western blot.

This is a sound suggestion. Unfortunately, our methods exclude this possibility. We performed our knock-down experiments on dispersed primary pituitary cell suspension. In addition, we found that only 30-40% of the cells on each coverslip were successfully transduced (virally), as assessed by the presence of enhanced green fluorescent protein. Together it would unfortunately not be possible to perform western blot analysis.

In addition, our electrophysiology experiments (dynamic clamp, current and voltage clamp) strongly suggest that in the primary pituitary cells only a small fraction of NALCN channels needs to be absent (knocked down) to produce such profound suppressive electrical effects on the cells (also alluded to by reviewer 1's comment, point 5). It is most likely therefore that even if western blot could be performed it would not be sensitive enough to meaningfully capture such small variation in channel fraction, especially given the low level of biological materials available.

2. In Figure 1, a negative control should be shown parallel to A and indicates where the H is taken.

We thank the reviewer for this comment, and we are happy to report that the specificity of this antibody against NALCN used in our study has previously been validated by previous studies, both through pre-absorption and in conditional knockout mice for NALCN (Zhang et al., 2021; Li et al., 2021). We apologise for not providing the information regarding this in our manuscript and as such, to reflect this we have added the following to our results section "*that is specific to the NALCN protein (Zhang et al., 2021; Li et al., 2021)*", (Page 9, lines 346 – 347).

In addition, we also provide here a figure from two experimental replicates (please see figure 1 below – same figure shown in response to reviewer 1's point 2 above) to show NALCN immunofluorescence (NALCN-IF) in HEK293T cells following transfection of NALCN-GFP along with its subunits (NALCN+UNC-79+UNC-80+FAM155A) (provided by our co-author Arnaud Monteil). We observed 1:1 colocalization of signals (NALCN-IF and NALCN-GFP).

Figure 1

With regards to panel H, based on the comments from both reviewers, we have decided to remove panel H, as it is not an appropriate control. Instead, we reference previous studies that have validated the NALCN antibody used in our study (please see our above response). Please note that here we used a different colour fluorophore to localise NALCN-primary antibody reaction as GFP reports successful NALCN transfection

3. On line 379, the subtitle is not necessary and the content under this subtitle (lines 380-392) should be combined with the content under the subtitle on line 394.

We agree and thank the reviewer for highlighting this. We have merged these contents as suggested.

4. The subtitle of Figure 3 A) should be like " A representative voltage trace of silent NALCN KD pituitary cells with firing activity being restored" because it is not just a representative voltage trace of silent NALCN KD pituitary cells.

We thank the reviewer for this suggestion, and we have now edited this title to reflect this.

5. For the y-axis label in Figure 5 A, it should be indicated it is a ratio of 340/380 before $[Ca^{2+}]_i$ or concentration unit after $[Ca^{2+}]_i$.

We thank the reviewer for highlighting this mistake, and we have now changed this axis label to F340/380 $[Ca^{2+}]_i$ ratio.

6. On line 646 - 647, "low" should be added before $[Ca^{2+}]_e$ -induced depolarization

Thank you. We have now corrected this (Page 17, line 584).

References:

Li, J. et al. (2021) "Elevated expression and activity of sodium leak channel contributes to neuronal sensitization of inflammatory pain in rats," *Frontiers in Molecular Neuroscience*, 14. Available at: <https://doi.org/10.3389/fnmol.2021.723395>.

Zhang, D. et al. (2021) "Sodium leak channel contributes to neuronal sensitization in neuropathic pain," *Progress in Neurobiology*, 202, p. 102041. Available at: <https://doi.org/10.1016/j.pneurobio.2021.102041>.

Dear Dr Belle,

Re: JP-RP-2023-284036R1 "The background sodium leak channel NALCN is a major controlling factor in pituitary cell excitability" by Marziyeh Belal, Mariusz Mucha, Arnaud Monteil, Paul G Winyard, Robert Pawlak, Jamie J Walker, Joel Tabak, and Mino David Belle

Thank you for submitting your manuscript to The Journal of Physiology. It has been assessed by a Reviewing Editor and by 2 expert referees and we are pleased to tell you that it is potentially acceptable for publication following satisfactory major revision.

LANGUAGE EDITING AND SUPPORT FOR PUBLICATION: If you would like help with English language editing, or other article preparation support, Wiley Editing Services offers expert help, including English Language Editing, as well as translation, manuscript formatting, and figure formatting at www.wileyauthors.com/eoo/preparation. You can also find resources for Preparing Your Article for general guidance about writing and preparing your manuscript at www.wileyauthors.com/eoo/prepresources.

REVISION CHECKLIST:

We look forward to receiving your revised submission.

Yours sincerely,

Dr Peying Fong
Senior Editor
The Journal of Physiology
<https://jp.msubmit.net>
<http://jp.physoc.org>
The Physiological Society
Hodgkin Huxley House
30 Farringdon Lane
London, EC1R 3AW
UK
<http://www.physoc.org>
<http://journals.physoc.org>

REQUIRED ITEMS

- You must start the Methods section with a paragraph headed Ethical Approval. A detailed explanation of journal policy and regulations on animal experimentation is given in Principles and standards for reporting animal experiments in The Journal of Physiology and Experimental Physiology by David Grundy J Physiol, 593: 2547-2549. doi:10.1113/JP270818.). A checklist outlining these requirements and detailing the information that must be provided in the paper can be found at: <https://physoc.onlinelibrary.wiley.com/hub/animal-experiments>. Authors should confirm in their Methods section that their experiments were carried out according to the guidelines laid down by their institution's animal welfare committee, and conform to the principles and regulations as described in the Editorial by Grundy (2015). The Methods section must contain details of the anaesthetic regime: anaesthetic used, dose and route of administration and method of killing the experimental animals.
- You must start the Methods section with a paragraph headed Ethical Approval. If experiments were conducted on humans confirmation that informed consent was obtained, preferably in writing, that the studies conformed to the standards set by the latest revision of the Declaration of Helsinki, and that the procedures were approved by a properly constituted ethics committee, which should be named, must be included in the article file. If the research study was registered (clause 35 of the Declaration of Helsinki) the registration database should be indicated, otherwise the lack of registration should be noted as an exception (e.g. The study conformed to the standards set by the Declaration of Helsinki, except for registration in a database.). For further information see: <https://physoc.onlinelibrary.wiley.com/hub/human-experiments>.
- The Journal of Physiology funds authors of provisionally accepted papers to use the premium BioRender site to create high resolution schematic figures. Follow this link and enter your details and the manuscript number to create and download figures. Upload these as the figure files for your revised submission. If you choose not to take up this offer we require figures to be of similar quality and resolution. If you are opting out of this service to authors, state this in the Comments section on the Detailed Information page of the submission form. The link provided should only be used for the purposes of this submission. Authors will be charged for figures created on this premium BioRender account if they are not related to this manuscript submission.
- Please ensure that any tables are in Word format and are, wherever possible, embedded in the article file itself.
- Your paper contains Supporting Information of a type that we no longer publish. Any information essential to an understanding of the paper must be included as part of the main manuscript and figures. The only Supporting Information that we publish are video and audio, 3D structures, program codes and large data files. Your revised paper will be returned to you if it does not adhere to our Supporting Information Guidelines
- Papers must comply with the Statistics Policy https://jp.msubmit.net/cgi-bin/main.plex?form_type=display_requirements#statistics

In summary:

- If n {less than or equal to} 30, all data points must be plotted in the figure in a way that reveals their range and distribution.

A bar graph with data points overlaid, a box and whisker plot or a violin plot (preferably with data points included) are acceptable formats.

- If $n > 30$, then the entire raw dataset must be made available either as supporting information, or hosted on a not-for-profit repository e.g. FigShare, with access details provided in the manuscript.
- 'n' clearly defined (e.g. x cells from y slices in z animals) in the Methods. Authors should be mindful of pseudoreplication.
- All relevant 'n' values must be clearly stated in the main text, figures and tables, and the Statistical Summary Document (required upon revision).
- The most appropriate summary statistic (e.g. mean or median and standard deviation) must be used. Standard Error of the Mean (SEM) alone is not permitted.
- Exact p values must be stated. Authors must not use 'greater than' or 'less than'. Exact p values must be stated to three significant figures even when 'no statistical significance' is claimed.
- Statistics Summary Document completed appropriately upon revision.
- Please include an Abstract Figure file, as well as the figure legend text within the main article file. The Abstract Figure is a piece of artwork designed to give readers an immediate understanding of the research and should summarise the main conclusions. If possible, the image should be easily 'readable' from left to right or top to bottom. It should show the physiological relevance of the manuscript so readers can assess the importance and content of its findings. Abstract Figures should not merely recapitulate other figures in the manuscript. Please try to keep the diagram as simple as possible and without superfluous information that may distract from the main conclusion(s). Abstract Figures must be provided by authors no later than the revised manuscript stage and should be uploaded as a separate file during online submission labelled as File Type 'Abstract Figure'. Please ensure that you include the figure legend in the main article file. All Abstract Figures should be created using BioRender. Authors should use The Journal's premium BioRender account to export high-resolution images. Details on how to use and access the premium account are included as part of this email.

EDITOR COMMENTS

Reviewing Editor:

This paper is interesting but it is concerning that in many cases the data shown does not match the primary data. This needs to be fixed, before the paper can be accepted. Authors can be given a final chance to rectify the following issues and if it is still not acceptable, I would recommend a final rejection. I would not want to edit another round of revisions for this paper as we will exhaust our reviewers.

The major points to be fixed are below.

1. The authors should mention the caveat that the specificity of the NALCN Ab is not clearly established.
2. They need to show the data for the second shRNA-2 (that they claim to have done) as a supplemental figure.
3. They must include a comment on theoretical V_m depolarization achieved with 15 mM K^+ estimated with the Nernst equation as mentioned in the rebuttal letter.

They must also correct the following:

1. The primary data they show for Figure panel 3D is incorrect and must be corrected.
2. Verify whether the scale in Panel 5B the same as in Panel 5A. If not, add the scale in panel 5B or show both graphs at the same scale.
3. Primary data in panel 5C does not match that in the Figure. Please correct in case the primary data is incorrectly provided.
4. The primary data in Figure 6B is also problematic. Comment on why this inward current was removed? The figure should match the primary data.
5. Provided primary data for Figure panel 6D should matches what is shown in both Figures and Figure Legend 6D.

Senior Editor:

Your revised manuscript has been reviewed by the original Referees and Reviewing Editor. Both Referees appreciate its

overall potential for impact, but important concerns remain. Although Referee 2 overall is satisfied with your responses to initial queries, Referee 1 remains critical and also identifies additional concerns, primarily pertinent to data presentation errors and/or inaccuracies that are summarized by the Reviewing Editor. Please address all of Referee 1's points completely, within your revised manuscript. In particular, Referee 1 challenges your chosen approach toward antibody validation and proposes a straightforward and overall more robust option entailing an overexpression strategy. In this day of Rigor and Reproducibility initiatives that raise awareness regarding antibody specificity, peptide blocking approaches are considered at best to be a bare minimum for validation. In addition, incorporation of data acquired using a second shRNA, within Figure 2 proper and accompanied by appropriate narration in the Results section, would strengthen the manuscript and ensure the observation is robust and defensible. Also, if proper membrane potential measurements cannot be performed, then please include within the manuscript the theoretical membrane potential prediction as outlined in the rebuttal. Finally, fix the errors in data presentation flagged by Referee 1.

From the Information for Authors, note that a Statistical Summary Document is required for revised manuscripts. It seems a Data Summary Set was submitted for presentation as online Supporting Information. While this might be a misunderstanding, the Authors should be aware that a Data Summary Set/Supporting Information Document is not the same as a Statistical Summary Document. Please do provide the required Statistical Summary Document when submitting your revised manuscript.

REFEREE COMMENTS

Referee #1:

I appreciate the author's effort to address my previous comments/concerns I have on the manuscript. There are still some additional comments that have to be taken care of by the authors to make the paper acceptable for publication. In my honest opinion, I think the paper is very interesting and relevant to the field so please consider these additional comments:

1. Although the authors discuss previous publications showing the specificity of the NALCN Ab, the specificity of such antibody in those publications is also not clearly demonstrated since their knockdown cells still show Ab staining and only peptide blockade abolishes the signal detection by Ab. In my humble opinion, consider this method is not appropriate for Ab specificity since the peptide will of course be recognized by the Abs and block the staining. The overexpression of NALCN in HEK cells with the complex shows some specificity on GFP+ cells which could be a good control experiment so I will suggest the authors to include such data in the manuscript as a supplement figure if the data has not been previously published by their collaborator.
2. I appreciate the fact the authors have validated their original findings with a second shRNA which shows similar results but I don't understand why such valuable data is not included on the revised manuscript and it is only mentioned in the methods section as not shown. Since this shRNA-2 is mouse specific I will encourage the authors to include this new data, at least as a supplemental figure, to support their conclusions. Please provide primary data as well.
3. If the authors have not measured V_m upon 15 mM K^+ to induce depolarization-mediated calcium influx, at least include a comment on theoretical V_m depolarization achieved with 15 mM K^+ estimated with the Nernst equation as they have done in the rebuttal letter.

After careful revision of the primary data provided by the authors, there are additional major concerns that have to be taken care of, which includes:

1. Provided primary data for Figure panel 3D is incorrect. The authors have provided primary data for added conductance but the Figure 3D shows subtracted conductance. Please correct.
2. Provided primary data for Figure panel 5B shows values of calcium influx going over 0.5 ratio units (e.g. 0.517309) but the peak shown in Figure 5B reached its maximum below 0.5. Is the scale in Panel 5B the same as in Panel 5A. If not, please add the scale in panel 5B or put both graphs at the same scale. In the same Figure, primary data in panel 5C does not match what is shown on the Figure and indeed the differences are not significant. Please correct in case the primary data that has been provided is incorrect.
3. The primary data shown in Figure 6B shows a strong increase inward current in NALCN KD group similar to the NS control around 692.9s (-19.043 pA), but the Figure does not show that. Has this figure been altered? Please comment on why this inward current was removed? Was this an artifact? I understand this happens from switching from b to c steps but the figure should not be modified and it should match the primary data.
4. Provided primary data for Figure panel 6D includes only the average of delta current density but the Figure shows individual values and figure legend indicates delta median values. Please provide primary data that matches what is shown in both Figures and Figure Legend 6D.

Referee #2:

The authors have addressed my critiques very well.

END OF COMMENTS

1st Confidential Review

05-May-2023

Dear Editors,

We thank you for your continuing support of our manuscript.

We also thank the reviewers for their comments and time, and are happy to learn that we have fully addressed reviewer 2's comments. We are very grateful to reviewer 1 for the additional comments raised and for catching some of our oversights.

We have addressed those below on a point-by-point basis and have accordingly revised our manuscript. We have carefully gone through the manuscript, supporting materials, and primary data to ensure no further mistakes remained.

As requested, we are providing two versions of the manuscript. One with all relevant changes made to the document, based on this latest review, highlighted with blue text. The second is a clean version with no changes tracked.

We trust that our revisions sufficiently addressed all your and the reviewer's concerns.

Our thanks and best wishes,

Mino Belle

EDITOR COMMENTS

Reviewing Editor:

This paper is interesting but it is concerning that in many cases the data shown does not match the primary data. This needs to be fixed, before the paper can be accepted.

We sincerely apologize for these oversights when providing the primary data, and this has now been fully rectified. Please see below. In addition, we have carefully checked all primary data provided and the manuscript to ensure no mistakes remained.

Authors can be given a final chance to rectify the following issues and if it is still not acceptable, I would recommend a final rejection. I would not want to edit another round of revisions for this paper as we will exhaust our reviewers.

We are very grateful to the editors and reviewer.

The major points to be fixed are below.

1. The authors should mention the caveat that the specificity of the NALCN Ab is not clearly established.

We thank the editor for this suggestion and agree. We have now included a statement in our manuscript to say that the specificity of the NALCN Ab is not clearly established (Pages 9-10; last paragraph and in the legend of figure 1).

In addition, based on reviewer 2's suggestion, we have added a supplement figure (Supporting information, Fig. S1) showing our transfection control experiment in HEK293T cells. This is also appropriately referenced in the manuscript.

2. They need to show the data for the second shRNA-2 (that they claim to have done) as a supplemental figure.

We completely agree and based on the editor's comment below (paragraph from the senior editor) we have now incorporated this in the main Figures (proper) and Supporting information, Fig. S2. For not disrupting the flow of the manuscript (results), we have appropriately spread the panels into Figures 2, 3 and Supplemental figure 2 (specifically now as Fig. 2G; see also Figs, 3E and F; and Supporting information, Fig. S2 for respective experiments). We have accompanied this by appropriate narration in the Methods and Results. We have also provided primary data as requested by reviewer 1.

3. They must include a comment on theoretical Vm depolarization achieved with 15 mM K+ estimated with the Nernst equation as mentioned in the rebuttal letter.

We thank the editor, and this is now included in the manuscript (Pages 17-18, and in the legend of figure 5).

They must also correct the following:

1. The primary data they show for Figure panel 3D is incorrect and must be corrected.

We apologize for this oversight and thank the editor and reviewer for highlighting this. We have changed these to positive values. This is now the same as what is shown in figure panel 3D.

2. Verify whether the scale in Panel 5B the same as in Panel 5A. If not, add the scale in panel 5B or show both graphs at the same scale.

We again apologize for this oversight, and this has now been rectified. Due to similarities in the calcium measurements in NALCN KD cells and in their response to 15 mM K+, we accidentally picked datapoints from a different cell to include as primary data trace for Figure 5B. We have now replaced this with the exact trace shown as Figure 5B in the manuscript. We again apologize for this mistake and has included a sentence at the end of this figure legend to state that the Y-axis scale in panel 5A also applied to panel 5B, for clarity.

3. Primary data in panel 5C does not match that in the Figure. Please correct in case the primary data is incorrectly provided.

The primary data provided was incorrect and this has now been corrected.

4. The primary data in Figure 6B is also problematic. Comment on why this inward current was

removed? The figure should match the primary data.

What appeared to be an inward current was an artifact from switching from B to C steps, as identified by the reviewer. We removed this artifact for presentation purposes but have now provided the trace in its entirety in figure 6B to match what we provided in the primary data. We have commented in the figure legend that this is an artifact, to inform the reader.

5. Provided primary data for Figure panel 6D should matches what is shown in both Figures and Figure Legend 6D.

We again apologize for this oversight; the provided values were indeed the current values in pA unit and not the normalised values which is pA/pF; we have now provided the normalised values to match the primary data with what is shown in figure panel 6D.

Senior Editor:

Your revised manuscript has been reviewed by the original Referees and Reviewing Editor. Both Referees appreciate its overall potential for impact, but important concerns remain. Although Referee 2 overall is satisfied with your responses to initial queries, Referee 1 remains critical and also identifies additional concerns, primarily pertinent to data presentation errors and/or inaccuracies that are summarized by the Reviewing Editor. Please address all of Referee 1's points completely, within your revised manuscript. In particular, Referee 1 challenges your chosen approach toward antibody validation and proposes a straightforward and overall more robust option entailing an overexpression strategy. In this day of Rigor and Reproducibility initiatives that raise awareness regarding antibody specificity, peptide blocking approaches are considered at best to be a bare minimum for validation. In addition, incorporation of data acquired using a second shRNA, within Figure 2 proper and accompanied by appropriate narration in the Results section, would strengthen the manuscript and ensure the observation is robust and defensible. Also, if proper membrane potential measurements cannot be performed, then please include within the manuscript the theoretical membrane potential prediction as outlined in the rebuttal. Finally, fix the errors in data presentation flagged by Referee 1.

We thank the editors and have addressed all of Reviewer 1's points completely within our revised manuscript. Please see our point-by-point response below. Any alterations in the manuscript are highlighted in blue.

We have incorporated the data acquired using a second shRNA, within the main figures (proper) and Supporting information, Fig. S2. As mentioned above, for not disrupting the flow of the manuscript (results), we have appropriately spread the panels into Figures 2, 3 and Supporting information, Fig. S2 (specifically now as Fig. 2G; see also Figs, 3E and F; and Supporting information, Fig. S2 for respective experiments). We have accompanied this by appropriate narration in the Methods and Results. We have also provided primary data as requested by reviewer 1. In addition, based on reviewer 2's suggestion, we have added a supplement figure (Supporting information, Fig. S1) showing our transfection control experiment in HEK293T cells. This is also appropriately referenced

in the manuscript. Theoretical membrane potential is provided, and we have addressed all errors with regards to data presentation.

From the Information for Authors, note that a Statistical Summary Document is required for revised manuscripts. It seems a Data Summary Set was submitted for presentation as online Supporting Information. While this might be a misunderstanding, the Authors should be aware that a Data Summary Set/Supporting Information Document is not the same as a Statistical Summary Document. Please do provide the required Statistical Summary Document when submitting your revised manuscript.

We apologize and this has been a misunderstanding. We have ensured the Statistical Summary Document (downloaded from the Journal of Physiology website before July 2023, and completed) and other documents are uploaded through the correct online portals. The Statistical Summary Document/table is uploaded in the portal labelled as "Statistical Summary Document". We have uploaded the primary dataset excel file and Supplementary word document in the portal labelled as "Supporting Information for Online publication".

REFeree COMMENTS

Referee #1:

I appreciate the author's effort to address my previous comments/concerns I have on the manuscript. There are still some additional comments that have to be taken care of by the authors to make the paper acceptable for publication. In my honest opinion, I think the paper is very interesting and relevant to the field so please consider these additional comments:

We are very grateful to you for your comments and for identifying some major oversights in our manuscript and primary data. Thank you for your continuing support.

1. Although the authors discuss previous publications showing the specificity of the NALCN Ab, the specificity of such antibody in those publications is also not clearly demonstrated since their knockdown cells still show Ab staining and only peptide blockade abolishes the signal detection by Ab. In my humble opinion, consider this method is not appropriate for Ab specificity since the peptide will of course be recognized by the Abs and block the staining. The overexpression of NALCN in HEK cells with the complex shows some specificity on GFP+ cells which could be a good control experiment so I will suggest the authors to include such data in the manuscript as a supplement figure if the data has not been previously published by their collaborator.

Looking closely at the publications we agree, and thank the reviewer for the suggestion. We have now added this data in our manuscript as (Supporting information, Fig. S1) as it is still unpublished. We have modified the supporting text in the results to reflect both the reviewer's and editor's comments (Pages 9-13, in text and legend, all highlighted in blue text).

We apologize for not having included a supplement figure for this in our previous response. We assumed that the Journal of Physiology was still not accepting supplemental figures, thus provided

this information in our response letter for publication alongside our manuscript, if accepted. Happy that J. Physiol is now accepting supplement figures.

2. I appreciate the fact the authors have validated their original findings with a second shRNA which shows similar results but I don't understand why such valuable data is not included on the revised manuscript and it is only mentioned in the methods section as not shown. Since this shRNA-2 is mouse specific I will encourage the authors to include this new data, at least as a supplemental figure, to support their conclusions. Please provide primary data as well.

We thank the reviewer for this good recommendation, and we have now incorporated this in our main figures, as requested by the editor, and supplemental figure 2. We have also provided primary data as the reviewer requested.

For not disrupting the flow of the manuscript (results), we have appropriately spread the panels into Figures 2, 3 and Supporting information, Fig. S2 (specifically now as Fig. 2G; see also Figs, 3E and F, and Supporting information, Fig. S2 for respective experiments). We have accompanied this by appropriate narration in the Methods and Results.

3. If the authors have not measured Vm upon 15 mM K⁺ to induce depolarization-mediated calcium influx, at least include a comment on theoretical Vm depolarization achieved with 15 mM K⁺ estimated with the Nernst equation as they have done in the rebuttal letter.

This is a good suggestion and we have now provided this information in the manuscript (Pages 17 and 18, main text and legend).

After careful revision of the primary data provided by the authors, there are additional major concerns that have to be taken care of, which includes:

We deeply apologize for these oversights, and we have carefully corrected all. As mentioned, we are very grateful to the reviewer for highlighting these. Please see below.

1. Provided primary data for Figure panel 3D is incorrect. The authors have provided primary data for added conductance but the Figure 3D shows subtracted conductance. Please correct.

We have revised this for consistency, both in primary data and figure panel 3D.

2. Provided primary data for Figure panel 5B shows values of calcium influx going over 0.5 ratio units (e.g. 0.517309) but the peak shown in Figure 5B reached its maximum below 0.5. Is the scale in Panel 5B the same as in Panel 5A. If not, please add the scale in panel 5B or put both graphs at the same scale.

We have now rectified this. Due to similarities in the calcium measurements in NALCN KD cells and in their response to 15 mM K⁺, we accidentally picked data from a different cell to show as primary data for Figure 5B. We have now replaced this with the correct and exact trace as shown in Figure 5B in the manuscript. The scale in panel 5B is indeed the same as panel 5A. For clarity, we have included a sentence at the end of this figure legend to state that the Y-axis scale in panel 5A also applies to panel 5B.

In the same Figure, primary data in panel 5C does not match what is shown on the Figure and indeed the differences are not significant. Please correct in case the primary data that has been provided is incorrect.

Thank you and we have now provided the correct primary data set for this.

3. The primary data shown in Figure 6B shows a strong increase inward current in NALCN KD group similar to the NS control around 692.9s (-19.043 pA), but the Figure does not show that. Has this figure been altered? Please comment on why this inward current was removed? Was this an artifact? I understand this happens from switching from b to c steps but the figure should not be modified and it should match the primary data.

Thank you for identifying this. Indeed, this is an artifact from switching from B to C steps, as you rightly identified. We removed this artifact for presentation purposes but have now provided the trace in its entirety in figure 6B to match what we provide in the primary data for figure 6B. We have also commented on this in the figure legend to inform the reader.

4. Provided primary data for Figure panel 6D includes only the average of delta current density but the Figure shows individual values and figure legend indicates delta median values. Please provide primary data that matches what is shown in both Figures and Figure Legend 6D.

We again apologize for this, and have now provided primary data that matches what is shown in the figure panel 6D, figure, and figure legend.

In addition, during our revisions, we have also added a new reference, a review article (Monteil et al, 2024); page 4, and identified and corrected some other minor misreporting of values in the manuscript (to match what is in our primary data). All highlighted in blue. Please note that those do not change any of our results or conclusions.

Dear Dr Belle,

Re: JP-RP-2024-284036R2 "The background sodium leak channel NALCN is a major controlling factor in pituitary cell excitability" by Marziyeh Belal, Mariusz Mucha, Arnaud Monteil, Paul G Winyard, Robert Pawlak, Jamie J Walker, Joel Tabak, and Mino David Belle

Thank you for submitting your manuscript to The Journal of Physiology. It has been assessed by a Reviewing Editor and by 1 expert referee and we are pleased to tell you that it is acceptable for publication following satisfactory minor revision.

REVISION CHECKLIST:

- 'Potential Cover Art' for consideration as the issue's cover image

- Appropriate Supporting Information (Video, audio or data set: see https://jp.msubmit.net/cgi-bin/main.plex?form_type=display_requirements#supp).

We look forward to receiving your revised submission.

Yours sincerely,

Peying Fong
Senior Editor
The Journal of Physiology

REQUIRED ITEMS

- Your paper contains Supporting Information of a type that we no longer publish, including supplementary tables and figures. Any information essential to an understanding of the paper must be included as part of the main manuscript and figures. The only Supporting Information that we publish are video and audio, 3D structures, program codes and large data files. Your revised paper will be returned to you if it does not adhere to our Supporting Information Guidelines.

- Please provide a legend to accompany your abstract figure. This should be included in the main article (Word) file, along with the other figure legends.

- Please include your ethics committee approval reference number in the Methods section.

EDITOR COMMENTS

Reviewing Editor:

All points raised by my reviewers are now addressed.

Two Supplementary Figures should be included in the main manuscript.

Senior Editor:

In the review comments provided, you will see that your extra effort in addressing concerns raised in the previous round of review is appreciated. At this time, the referee, the Reviewing Editor, and I concur that your study carries potential to extend the collective understanding of pituitary cell excitability, and the contributions of sodium leak by NALCN.

There remains, however, an important matter of form that requires your attention. Both the Reviewing Editor and I note that Supplementary Figures were added de novo in a Supporting Information file accompanying your revised manuscript. According to the guidelines set forth in the Information for Authors, Supplemental Figures are not permitted. Please incorporate these images and plots into figures (1 and 6) for presentation within the manuscript proper.

The referee offers a few minor suggestions for your consideration. I encourage you to include acknowledgement information within the Acknowledgements section proper.

In addition, several other details require attention. Please include 1) the ethics approval number in the Methods and 2) a legend for the graphical abstract figure.

I look forward to receiving these revisions, and thank you for submitting your work to The Journal of Physiology.

REFEREE COMMENTS

Referee #1:

I would like to thank the authors for successfully addressing all my comments and concerns.

I have a couple of minor comments I recommend the editors to consider:

1. Delete blue text in Figure legend 1 (line 374-376) as it is redundant to the text in the result section (line 351-353).
2. Move the acknowledgements in Supplemental Figure legend 1 to the Acknowledgements section. In the same figure, line 1061, please correct "data not showed" by "data not shown".

END OF COMMENTS

2nd Confidential Review

23-Jul-2024

Dear Editors,

Thank you for your continuing support of our manuscript.

We are very pleased to learn that it is acceptable for publication following satisfactory minor revision, which we have now completed.

We have addressed those below on a point-by-point basis and have accordingly revised our manuscript.

As requested, we are providing two versions of the manuscript. One with all relevant changes made to the document, based on this latest review, highlighted with blue text. The second is a clean version with no changes tracked.

With thanks,

Mino Belle

REQUIRED ITEMS

- Your paper contains Supporting Information of a type that we no longer publish, including supplementary tables and figures. Any information essential to an understanding of the paper must be included as part of the main manuscript and figures. The only Supporting Information that we publish are video and audio, 3D structures, program codes and large data files. Your revised paper will be returned to you if it does not adhere to our [Supporting Information Guidelines \[jp.msubmit.net\]](http://jp.msubmit.net).

Thank you and as requested, we have now incorporated images and plots from our supplementary figures into figures (1 and 6) for presentation within the manuscript proper. We have updated the "Primary Dataset" document to reflect this for Figure 6. For Figure 1, please note that we have removed Figure 1I (as it was a replicate of Figure 1H) and replaced Figure 1H with higher resolution images. We have mentioned in the figure legend that two replicates were performed revealing similar results. We hope that this meets with your approval.

- Please provide a legend to accompany your abstract figure. This should be included in the main article (Word) file, along with the other figure legends.

This is now provided – placed at the end of the main article file alongside the abstract figure (after the references).

- Please include your ethics committee approval reference number in the Methods section.

This is now provided (Page 5; line 177).

EDITOR COMMENTS

Reviewing Editor:

All points raised by my reviewers are now addressed.

Two Supplementary Figures should be included in the main manuscript.

Completed. Please see our above response.

Senior Editor:

In the review comments provided, you will see that your extra effort in addressing concerns raised in the previous round of review is appreciated. At this time, the referee, the Reviewing Editor, and I concur that your study carries potential to extend the collective understanding of pituitary cell excitability, and the contributions of sodium leak by NALCN.

There remains, however, an important matter of form that requires your attention. Both the Reviewing Editor and I note that Supplementary Figures were added de novo in a Supporting Information file accompanying your revised manuscript. According to the guidelines set forth in the Information for Authors, Supplemental Figures are not permitted. Please incorporate these images and plots into figures (1 and 6) for presentation within the manuscript proper.

We are pleased to learn this and as requested we have incorporated images and plots from our supplementary figures into figures (1 and 6) for presentation within the manuscript proper. Please note that we have used higher resolution images in Figure 1H.

The referee offers a few minor suggestions for your consideration. I encourage you to include acknowledgement information within the Acknowledgements section proper.

We agree and this is now done.

In addition, several other details require attention. Please include 1) the ethics approval number in the Methods and 2) a legend for the graphical abstract figure.

Both points 1) and 2) are now done.

I look forward to receiving these revisions, and thank you for submitting your work to The Journal of Physiology.

REFEREE COMMENTS

Referee #1:

I would like to thank the authors for successfully addressing all my comments and concerns.

Thank you for your support and rigor.

I have a couple of minor comments I recommend the editors to consider:

1. Delete blue text in Figure legend 1 (line 374-376) as it is redundant to the text in the result section (line 351-353).

This is now deleted.

2. Move the acknowledgements in Supplemental Figure legend 1 to the Acknowledgements section.

This is now done.

In the same figure, line 1061, please correct "data not showed" by "data not shown".

Corrected.

Dear Dr Belle,

Re: JP-RP-2024-284036R3 "The background sodium leak channel NALCN is a major controlling factor in pituitary cell excitability" by Marziyeh Belal, Mariusz Mucha, Arnaud Monteil, Paul G Winyard, Robert Pawlak, Jamie J Walker, Joel Tabak, and Mino David Belle

Thank you for submitting your manuscript to The Journal of Physiology. It has been assessed by an Editor and we are pleased to tell you that it is acceptable for publication following satisfactory minor revision.

REVISION CHECKLIST:

Please upload two versions of your manuscript text: one with all relevant changes highlighted and one clean version with no changes tracked. The manuscript file should include all tables and figure legends, but each figure/graph should be uploaded as separate, high-resolution files. The journal is now integrated with Wiley's Image Checking service. For further details, see: <https://www.wiley.com/en-us/network/publishing/research-publishing/trending-stories/upholding-image-integrity-wileys->

image-screening-service

We look forward to receiving your revised submission.

Yours sincerely,

Peying Fong
Senior Editor
The Journal of Physiology

EDITOR COMMENTS

Thanks very much for responding to comments of the previous review cycle. It is much improved.

However, I do have one remaining suggestion (below).

The point is stylistic and concerns the referencing of Chua et al, 2020 in the Acknowledgement. It appears within parentheses and specifically pertains to Methods applied and reagents obtained for control experiments performed for figure 1H. This figure panel was incorporated into the manuscript proper in response to comments requesting Supporting Information be fully integrated. I note the previous review requested adjustments in the Acknowledgements as well. The Acknowledgements is an unconventional place for such a reference, and it prompted me to review the Methods section further. Although I do note the control experiment has been written descriptively into the Results section, specific details (HEK293T cell culture, transfection, reagents used, staining, microscopy, any analyses procedure for co-localization if applicable, etc. as well as reference to Chua et al, 2020 for providing the plasmids transfected) properly should be provided in the Methods section.

END OF COMMENTS

3rd Confidential Review

05-Oct-2024

Dear Editor,

Thank you for your thoroughness. It is very much appreciated, and we have now provided these details in the methods section.

With thanks,

Mino Belle

EDITOR COMMENTS

Thanks very much for responding to comments of the previous review cycle. It is much improved.

However, I do have one remaining suggestion (below).

The point is stylistic and concerns the referencing of Chua et al, 2020 in the Acknowledgement. It appears within parentheses and specifically pertains to Methods applied and reagents obtained for control experiments performed for figure 1H. This figure panel was incorporated into the manuscript proper in response to comments requesting Supporting Information be fully integrated. I note the previous review requested adjustments in the Acknowledgements as well. The Acknowledgements is an unconventional place for such a reference, and it prompted me to review the Methods section further. Although I do note the control experiment has been written descriptively into the Results section, specific details (HEK293T cell culture, transfection, reagents used, staining, microscopy, any analyses procedure for co-localization if applicable, etc. as well as reference to Chua et al, 2020 for providing the plasmids transfected) properly should be provided in the Methods section.

Thank you, and all these information are now provided in the Methods section (pages 9 and 10).

Dear Dr Belle,

Re: JP-RP-2024-284036R4 "The background sodium leak channel NALCN is a major controlling factor in pituitary cell excitability" by Marziyeh Belal, Mariusz Mucha, Arnaud Monteil, Paul G Winyard, Robert Pawlak, Jamie J Walker, Joel Tabak, and Mino David Belle

We are pleased to tell you that your paper has been accepted for publication in The Journal of Physiology.

*****IMPORTANT*****

Thank you for providing clarification in relation to our recent image check query. We appreciate your cooperation with this process.

Please can you confirm with Diana at the Editorial Office - is it just the legend for Figure 1 that needs to be updated before sending on to production? (and the figure file itself is OK?).

Please email Diana at: jp@physoc.org

Many thanks!

Yours sincerely,

Peying Fong
Senior Editor
The Journal of Physiology

If you would like to receive our 'Research Roundup', a monthly newsletter highlighting the cutting-edge research published in The Physiological Society's family of journals (The Journal of Physiology, Experimental Physiology, Physiological Reports, The Journal of Nutritional Physiology and The Journal of Precision Medicine: Health and Disease), please click this link, fill in your name and email address and select 'Research Roundup':
<https://www.physoc.org/journals-and-media/membernews>

• **TRANSPARENT PEER REVIEW POLICY:** To improve the transparency of its peer review process, The Journal of Physiology publishes online as supporting information the peer review history of all articles accepted for publication. Readers will have access to decision letters, including Editors' comments and referee reports, for each version of the manuscript as well as any author responses to peer review comments. Referees can decide whether or not they wish to be named on the peer review history document.

• You can help your research get the attention it deserves! Check out Wiley's free Promotion Guide for best-practice recommendations for promoting your work at: www.wileyauthors.com/eeo/guide. You can learn more about Wiley Editing Services which offers professional video, design, and writing services to create shareable video abstracts, infographics, conference posters, lay summaries, and research news stories for your research at: www.wileyauthors.com/eeo/promotion.

• **IMPORTANT NOTICE ABOUT OPEN ACCESS:** To assist authors whose funding agencies mandate public access to published research findings sooner than 12 months after publication, The Journal of Physiology allows authors to pay an Open Access (OA) fee to have their papers made freely available immediately on publication.

The Corresponding Author will receive an email from Wiley with details on how to register or log-in to Wiley Authors Services where you will be able to place an order

• You can check if your funder or institution has a Wiley Open Access Account here: <https://authorservices.wiley.com/author->

<resources/Journal-Authors/licensing-and-open-access/open-access/author-compliance-tool.html>.

EDITOR COMMENTS

All remaining points have been addressed satisfactorily, and this manuscript is now ready for final acceptance. Congratulations!

Thank you for favoring The Journal of Physiology with this interesting and important study.